

# A novel isotope pool dilution approach to quantify gross rates of key abiotic and biological processes in the soil phosphorus cycle

Wolfgang Wanek*, David Zezula, Daniel Wasner, Maria Mooshammer, Judith Prommer

Division of Terrestrial Ecosystem Research, Department of Microbiology and Ecosystem Science, Research Network „Chemistry meets Microbiology", University of Vienna, Althanstraße 14, 1090 Vienna, Austria

*Correspondence to*: Wolfgang Wanek (wolfgang.wanek@univie.ac.at)

**Abstract.**

Efforts to understand and model the current and future behavior of the global phosphorus (P) cycle are limited by the availability of global data on gross rates of soil P processes, as well as its environmental controls. We here present a novel isotope pool dilution approach using $^{33}P$ labelling of live and sterile soils, which allows to obtain high quality data on gross fluxes of soil inorganic P ($P_i$) sorption and desorption, as well as of gross fluxes of organic P mineralization and microbial $P_i$ uptake. At the same time, net immobilization of $^{33}P_i$ by soil microbes and abiotic sorption can be easily derived and partitioned. Compared to other approaches, we used short incubation times (up to 48 h), avoiding tracer re-mineralization, which was confirmed by separation of organic P and $P_i$ using isobutanol fractionation. This approach is also suitable for strongly weathered and P impoverished soils, as sensitivity is increased by extraction of exchangeable bio-available $P_i$ (Olsen $P_i$; 0.5 M $NaHCO_3$) followed by $P_i$ measurement using the malachite green assay. Biotic processes were corrected for desorption/sorption processes by using adequate sterile abiotic controls that exhibited negligible microbial and extracellular phosphatase activities. Gross rates are calculated using analytical solutions of tracer kinetics, which also allows to study gross soil P dynamics under non-steady-state conditions. Finally, we present major environmental controls of gross and net P cycle processes that were measured for three P-poor tropical forest and three P-rich temperate grassland soils.

**Keywords**: phosphorus, organic P mineralization, sorption, desorption, isotope pool dilution, $^{33}P$;

## 1   Introduction

Phosphorus (P) is a major limiting nutrient to terrestrial primary production, particularly so on highly weathered soils, as, e.g. found in the tropics. Globally, increasing imbalances between nitrogen (N) and P inputs (i.e.

increasing N:P stoichiometry of inputs) caused by human activities and land-use changes through increased emissions of reactive N is suggested to lead to progressive P limitation of terrestrial ecosystems, and first signs thereof have been identified (Penuelas et al., 2013). A decrease in the relative P availability might have strong repercussions on future nutrient limitations of natural ecosystems, food production and carbon (C) sequestration (Penuelas et al., 2013;Penuelas et al., 2012;Yang et al., 2013). Efforts to understand and model the current and

future global P cycle and its coupling to the global C and N cycles have been intensified, but are strongly limited





by the availability of global data on soil gross P processes and their environmental controls (Reed et al., 2015). Large investments into new projects, experiments and models have therefore been recently undertaken to advance our understanding of the terrestrial P cycle, and to fill data gaps, e.g. IMBALANCE-P (http://imbalancep-erc.creaf.cat) and NGEE-TROPICS (http://ngee-tropics.lbl.gov).

40        Soil $P_i$ availability is governed by transfers between pools of exchangeable P, immobilized/fixed P and occluded P, by slow release of $P_i$ from mineral P through weathering of primary minerals, and by mineralization of organic P ($P_o$) (Fig. 1) (Bünemann, 2015;Turner et al., 2007). In strongly weathered soils, primary mineral P pools are depleted, and the largest fraction of P is found in occluded and fixed pools, as well as in $P_o$ (Vitousek and Farrington, 1997;Yang and Post, 2011). Phosphorus limitation in such soils is further aggravated by their high

P sorption potentials caused by high contents of Fe-Al (hydr)oxides (Goldberg and Sposito, 1985). Notably, microbial biomass P represents a sizeable P pool in soils compared to C and N, and can account for up to 40% of total soil $P_o$ and up to 70% of the total biomass P, including plants and soil microbes (Turner et al., 2013;Xu et al., 2013). Microbial biomass therefore represents an important buffer of bio-available P and is an important mediator of the terrestrial P cycle through $P_o$ mineralization and $P_i$ immobilization (Achat et al., 2012). Most of the

immediate P needs of plants (and microbes) in natural and agricultural systems is supplied by $P_o$ mineralization, catalyzed by extracellular phosphatases that are released by soil microbes and plant roots (Richardson and Simpson, 2011), as well as by abiotic $P_i$ desorption. Soil microbes and plant roots can also promote the release of P from primary and secondary minerals by accelerating abiotic processes, namely mineral dissolution and $P_i$ desorption, through exudation of (phyto)siderophores and organic acids (Mander et al., 2012;Ryan et al., 2001).

55        Soil P cycling processes such as soil $P_i$ sorption/desorption fluxes and gross $P_o$ mineralization rates, as well as the size of the exchangeable soil $P_i$ pool have most commonly been measured by isotope exchange (IEX) techniques, also termed isotope dilution (ID) approaches, using $^{32}P$ or $^{33}P$. These techniques are based on recurrent measurements of radiotracer recovery and $P_i$ concentration in soil water extracts (Di et al., 1997;Frossard et al., 2011;Bünemann, 2015) (Table 1). However, only during the last decade common, accepted protocols have become

adopted and are used to measure soil P processes following Oehl et al. (2001b). Nonetheless, a variety of other procedures and protocols is in use, and optimizations in methodology have been called for, particularly for $P_o$ mineralization (Bünemann, 2015). In short-term IEX experiments abiotic sorption/desorption processes from an isotopically exchangeable $P_i$ pool are measured as $E_{(t)}$-value over a short time period in batch experiments (100 min, 1:10 (w:v) soil: water slurry, ± microbicides), assuming that no microbial tracer uptake occurs, and

extrapolated to the total length of the main IEX incubation experiments (Table 1). Isotopic dilution ($E'_{(t)}$) is then measured over the full length of an incubation experiment lasting for several days to weeks, constituting the total amount of exchangeable $P_i$ or isotope dilution caused by concurrent biological processes ($P_o$ mineralization) and physicochemical processes. The difference between $E'_{(t)}$ and the extrapolated $E_{(t)}$-value provides then the measure of gross $P_o$ mineralization.

70       The isotope pool dilution approach (IPD) of Kirkham and Bartholomew (1954) also relies on the labelling of the $P_i$ pool with $^{33}P$ or $^{32}P$ and on subsequent time-resolved measurements of concentrations and specific activities of $P_i$ (Table 1, Figure 1B). However, in contrast to IEX techniques, changes in $P_i$ concentrations and specific activities are then solved by mass balance equations developed specifically for gross rate calculations based on tracer studies (Kirkham and Bartholomew, 1954). In the following we list the criteria that have to be met

by the IPD method to correctly determine gross rates of soil $P_o$ mineralization and soil $P_i$ sorption/desorption (Di et al., 2000;Murphy et al., 2003;Kirkham and Bartholomew, 1954).



1. The tracer ($^{32}P_i$ or $^{33}P_i$) and tracee (unlabeled $^{31}P_i$) behave identically and are well mixed. This is given for the different isotopes of P as long as radiotracer solution is homogeneously distributed in the soil and sufficient time is provided for isotope equilibration between added radiotracer and the native $P_i$ pool.

2. The influx into the target ($P_i$) pool (i.e. the product of $P_o$ mineralization) has to be unlabeled (i.e. no tracer remineralization), in order for it to dilute the tracer: tracee ratio over time (Figure 1B and C). Tracer remineralization via microbial tracer assimilation, mortality and subsequent remineralization of labelled $P_o$ would result in an underestimation of $P_o$ mineralization, but can be avoided by short incubation times (1-2 days).

3. Abiotic release of $P_i$ from a non-extractable pool ($P_i$ desorption) causes an influx of unlabeled $P_i$ into the target pool, resulting in an overestimation of the biotic process, $P_o$ mineralization, and has to be determined in parallel abiotic incubations of sterile soils. However, adequate abiotic controls with no contribution of biological processes has remained a major obstacle in measuring soil P dynamics with radiotracers, both in IEX/ID and IPD experiments. Procedures in earlier studies ranged from short-term assays with no inhibitor

addition as often performed in IEX assays (Spohn et al., 2013;Oehl et al., 2001b), to amendments of $HgCl_2$, sodium azide, toluene or chloroform, and gamma irradiation or repeated autoclaving (Kellogg et al., 2006;Bünemann, 2015;Bünemann et al., 2007;Oehl et al., 2001b;Achat et al., 2010).

4. The soil extraction should target the bio-available exchangeable $P_i$ pool. $P_i$ in soil solution, however, undergoes rapid equilibration with easily adsorbed $P_i$. An incomplete extraction of this pool causes an

underestimation of $P_o$ mineralization rates, due to desorption from this pool, causing an influx of unlabeled tracer (together with unlabeled $P_i$) into the target pool, and thus violates assumption #2 of IPD assays. The commonly used soil water extractions target only a small fraction of this target pool, whereas standard soil P extractants, such as Olsen, Mehlich-3 or Bray-1, extract a larger fraction (Kleinman et al., 2001) and, therefore, are suggested to be better suited to extract the rapidly exchanging $P_i$ pool (Kellogg et al., 2006).

5. The efflux from the isotopically labelled pool (i.e. microbial $P_i$ immobilization and $P_i$ sorption into a non-extractable pool) occurs at the ratio of tracer: tracee as present in the $P_i$ pool at any specific time, with no discrimination between native $P_i$ and added radiotracer (Figure 1B). A short pre-incubation time is therefore needed to allow for full mixing and isotopic equilibration of tracer and tracee (see point #1).

6. Changes in specific activity needs to be measured specifically in the target pool, i.e. in extractable $P_i$ for

measurements of gross rates of $P_o$ mineralization and $P_i$ sorption/desorption. However, most current approaches do not separate extractable $P_i$ and $P_o$ but measure radioactivity in unfractionated extracts, including radiolabeled $P_o$ formed during the incubation, leading to an eventual overestimation of $P_o$ mineralization.

7. The rates of $P_i$ influx ($P_o$ mineralization, abiotic $P_i$ release) and $P_i$ efflux (biotic and abiotic $P_i$ immobilization)

need to be constant over the duration of incubation: (i) the initial phase of fast immobilization by sorption, microbial uptake and isotopic equilibration of radiotracer is excluded from calculations of gross rates, and (ii) incubation takes place within a suitable timeframe to avoid microbial turnover and $^{33}P_o$ remineralization (see point #2). The two time points necessary to measure concentration and specific activity of $P_i$ for the IPD calculations should therefore lie in between the initial phase and the start of re-mineralization.

Mooshammer et al. (2012) adopted such a protocol for measurements of gross $P_o$ mineralization in decomposing plant litter, following the knowledge of IPD processes based on $^{15}N$ additions to study gross rates of soil N cycling (Hart et al., 1994;Murphy et al., 2003;Wanek et al., 2010;Braun et al., 2018). However, in plant litter P sorption



and the abiotic release of $P_i$ from sorbed P pools do not interfere. Consequently, the litter protocol cannot be directly transferred to soil studies. In the present study we developed an IPD protocol to assess soil P dynamics,

based on the previous work for litter by Mooshammer et al. (2012) and soils by Kellogg et al. (2006). The protocol is based on IPD theory (Kirkham and Bartholomew, 1954;Di et al., 2000) applied to parallel incubations of live and sterile soil with $^{33}P_i$ tracer addition. Gross rates of $P_i$ sorption (abiotic immobilization) and $P_i$ desorption are determined in sterile soils, and allow correction of gross $P_o$ mineralization and microbial $P_i$ immobilization rates in live soils. We used bicarbonate extractions to target the bio-available exchangeable $P_i$ pool. To avoid tracer re-

mineralization, we used short incubation periods (up to 2 days). To confirm that no significant amount of $^{33}P_o$ was formed during incubation, $P_i$ was also separated from $P_o$ based on isobutanol fractionation (Jayachandran et al., 1992). $P_i$ concentrations were measured based on the phosphomolybdate blue protocol, and at very low Pi concentrations, e.g. in tropical soils, that are below the detection limit of the phosphomolybdate blue method, were determined by parallel measurements of $P_i$ in bicarbonate extracts using the more sensitive malachite green assay

(D'Angelo et al., 2001;Ohno and Zibilske, 1991). The protocol was tested rigorously with two different soils, and then applied to in total six soils (tropical forest and temperate grassland) to explore environmental controls on gross soil P dynamics.

## 2   Materials and methods

### 2.1   Soil materials and basic characterization

Soils (0-15 cm depth) were collected in summer 2015 from three temperate grassland sites in Austria and in spring 2015 from three tropical lowland forest sites in Costa Rica (Table 2). The grassland soils were extensively managed meadows, collected in Lower Austria (48° 13-20' N, 16° 12-17' E) in the vicinity of Vienna, at elevations between 170 and 320 m. The tropical forest soils were collected along a topographic gradient (ridge-slope-valley bottom)

in wet evergreen old-growth forests in SW Costa Rica close to the National Park Piedras Blancas (8° 41' N, 83° 12' W, 110-250 m a.s.l.). Soils were sieved to 2 mm and stored in an air-dried state. Soil pH was measured in a 1:5 (w:v) mixture of air dried soil in water after 60 min of equilibration using an ISFET electrode (Sentron SI600 pH Meter). Soil texture was quantified using a miniaturized pipette/sieving protocol for 2-4 g air dried soils (Miller and Miller, 1987), using 4% sodium metaphosphate as a dispersant. Soil total C and total soil N content were

determined after grinding oven dried soil in a ball mill, using an elemental analyzer (EA 1110, CE Instruments, Thermo Scientific). Temperate grassland soils were treated with 2 M HCl to remove carbonates, re-dried, ground and then analyzed by elemental analyzer for soil organic C. Total soil P and total soil $P_i$ were measured after 0.5 M $H_2SO_4$ extraction of ignited soils (5 h at 450°C in a muffle furnace; (O´Halloran and Cade-Menun, 2008)) and of untreated soils, respectively, by the malachite green method (Ohno and Zibilske, 1991;D'Angelo et al., 2001).

Total organic P was estimated by calculating the difference between total soil P and total soil $P_i$.

### 2.2   Soil pre-treatment and assay of sterilization efficiency (abiotic controls)

Before starting the experiments, the soils were re-equilibrated from an air-dried state by rewetting to 60% water holding capacity for 6 days at 20°C. Gravimetric soil water content and water holding capacity were determined

prior to the experiment. Soils were then either sterilized twice, 48 and 2 h before start of the IPD experiments, by autoclaving at 121°C for 60 min, or were left at 20°C. Sterilization efficiency was checked based on soil enzyme activity measurements. Fluorescein diacetate (FDA) hydrolysis in soils was measured as a proxy of viable, active



microbial biomass (Green et al., 2006;Schnurer and Rosswall, 1982), and the activity of acid phosphomonoesterases, which are extracellular enzymes involved in $P_o$ mineralization, was determined using

methylumbelliferyl (MUF)-phosphate (Sirova et al., 2013;Marx et al., 2001).

### 2.3   $^{33}$P IPD assay

Duplicate soil aliquots (2 g fresh weight) of sterile and live soil were amended each with 20 kBq $^{33}P_i$ (dilution of orthophosphoric acid phosphorus-33 radionuclide, 5 mCi mL$^{-1}$, i.e. 185 MBq mL$^{-1}$ HCl-free water at specified

date, Perkin NEZ080002MC). Between 0.15-0.2 mL of $^{33}$P-label solution was added to each sample; the volume added was adjusted for each soil type to obtain an optimal water content in each soil (~75 % water holding capacity). Soils were extracted with 30 mL (temperate soils) or 15 mL (tropical soils) of 0.5 M NaHCO$_3$ (pH 8.5) after 4 and 24 h of incubation for 30 min on a horizontal shaker and filtered through ash-free cellulose filters.

Thereafter isobutanol fractionation of the bicarbonate extracts was performed, separating $P_i$ (into the

organic phase) from $P_o$ (into the acidic aqueous phase) allowing measurement of the kinetics and specific activity of the $P_i$ pool without interference of $P_o$ (Kellogg et al., 2006;Mooshammer et al., 2012). Isobutanol partitioning enables 100% recovery of $P_i$ with no hydrolysis of $P_o$ (Jayachandran et al., 1992). For isobutanol fractionation each 1.5 mL of soil extracts, standards and blanks were amended by sequential addition of 1.5 mL acidified molybdate, 3 mL deionized water and 3 mL isobutanol. The acidified molybdate reagent consists of 5 g ammonium

molybdate tetrahydrate ((NH$_4$)$_6$Mo$_7$O$_{24}$.4H$_2$O) dissolved in 0.1 L 2.3 M H$_2$SO$_4$ (stable at room temperature for at least three months) and causes strong CO$_2$ outgassing from the bicarbonate extracts. After addition of all reagents the vials were shaken overhead for 1 min and then rested for 10 min for phase separation. For later photometric quantification of $P_i$ in the isobutanol phase, standards ranging from 320 to ~1 μM $P_i$ (1:2 dilution series) and blanks, both of the same matrix as soil extracts (i.e. 0.5 M NaHCO$_3$), were prepared and underwent isobutanol

fractionation together with the samples. $^{33}$P recovery standards were also prepared and processed through the isobutanol fractionation protocol, consisting of the same volume of extractant (15 or 30 mL) and $^{33}$P tracer activity as added to soils.

$P_i$ in the isobutanol phase was quantified using the phosphomolybdate blue color reaction according to Murphy and Riley (1962). Briefly, each 1.5 mL of the upper organic phase were transferred to vials and amended

with 2.1 mL molybdate free reducing agent, consisting of 1.32 g ascorbic acid dissolved in 250 mL antimony potassium tartrate (APT) solution (145.4 mg APT in 0.5 M H$_2$SO$_4$). The APT solution is stable at room temperature for >4 weeks, whereas the molybdate free reducing agent has to be prepared fresh daily. Thereafter samples were shaken overhead for 1 min and rested for 20 min for phase separation and color development. A volume of 250 μL of the blue isobutanol phase was then pipetted into a microtiter plate and absorbance was read at 725 nm with

a microplate photometer (Tecan Infinite M200, Tecan Austria GmbH, Grödig, Austria).

In parallel to the phosphomolybdate blue reaction of $P_i$ in the isobutanol phase, $P_i$ concentrations were also determined directly in acidified bicarbonate extracts using the malachite green approach (D'Angelo et al., 2001). This method is 4-10 times more sensitive than the commonly used phosphomolybdate blue method and was chosen to account for the expectedly low $P_i$ concentrations of the tropical soils. Standards for calibration of the

malachite green method were prepared in 0.5 M NaHCO$_3$, ranging from 50 to 0.039 μM $P_i$. Acidification of bicarbonate extracts and standards (blanks) was performed on 2.5 mL sample aliquots by adding 250 μL 2.75 M H$_2$SO$_4$. Of the acidified samples and standards, 200 μL were pipetted into a microtiter plate, 40 μL malachite green reagent A were added and incubated for 10 min. Then 40 μL reagent B were added and absorbance was read after





45 min at 610 nm with a microplate reader. Reagent A was prepared by adding 50 mL deionized water in an amber
0.1 L glass bottle, adding 16.8 mL concentrated $H_2SO_4$, stirring and dissolving 1.76 g ammonium heptamolybdate
tetrahydrate $((NH_4)_6Mo_7O_{24} \cdot 4\ H_2O)$. Reagent B was prepared by heating 0.25 L of distilled $H_2O$ to 80°C in an
amber 0.5 L glass bottle, dissolving 0.875 g PVA (polyvinyl alcohol, MW = 72000 g/mol) whilst continuously
stirring, cooling to room temperature, and finally dissolving 87 mg malachite green oxalate in this solution. Both
reagents are stable for >6 months at room temperature.

Radioactivity ($^{33}P$ activity) was measured in 0.25 mL aliquots of acidified bicarbonate extracts and in 0.4
mL aliquots of the isobutanol phase, after addition of each 4 mL scintillation cocktail (Ultima Gold, Perkin Elmer),
by scintillation counting (Tri-Carb 1600 TR, Packard, Perkin Elmer).

### 2.4 Experiments

(i)      Time kinetics: high resolution time kinetics of tracer and tracee dynamics ($^{33}P_i$, $^{31}P_i$) were measured
in two soils (temperate grassland, soil 4; tropical forest, soil 3; Table 2), according to the procedure
as outlined above. After tracer addition to live and sterile soils in triplicates IPD assays were stopped
by extraction with 0.5 M $NaHCO_3$ after 0, 1, 2, 4, 8, 24, and 48 h.

(ii)      Microbial $^{33}P$: the above mentioned procedure can be combined with direct determination of
microbial P by extraction with liquid chloroform-enriched salt solutions (Setia et al., 2012). We here
tested a sequential extraction-liquid chloroform extraction (sECE) procedure. After 24 h of soil
incubation in experiment (i), soil samples (2 g fresh weight) were first extracted with 15 (soil 4) or
30 (soil 3) mL 0.5 M $NaHCO_3$ for 30 min, centrifuged for 15 min at 10.000 g, and the supernatant
was decanted. The soil residue was then re-extracted with 15 (30) mL 0.5 M $NaHCO_3$ containing 3%
(v:v) chloroform for 30 min and finally filtered through ash-free cellulose filters. Volume corrections
were applied for extractant absorption by the soil pellet after centrifugation.

(iii)      Soil effects on tracer dynamics: live and sterile soils (2 g aliquots) of all 6 soils (Table 2) were
measured in triplicates for $^{33}P_i$ activity and $P_i$ concentrations, and assays were stopped after 0, 4 and
24 h. Net immobilization of $^{33}P$ and gross process rates were calculated for the time interval 4 to 24
h, and relationships between gross and net soil P processes and soil physicochemical parameters were
tested.

### 2.5 Calculations of abiotic and biotic net $^{33}P$ immobilization

Additionally to the measurement of gross rates, abiotic net $^{33}P$ immobilization (net soil $P_i$ fixation) and biotic net
$^{33}P$ immobilization (net soil microbial $P_i$ immobilization) were calculated based on the determination of the
recovery of added tracer in soil extracts of live and autoclaved soils (see above). Abiotic immobilization (in %
added tracer) was estimated as 100 percent minus the percent $^{33}P$ recovery in autoclaved soils. Total
immobilization was estimated as 100 minus the percent $^{33}P$ recovery in live soils. Biotic immobilization was
calculated as the difference between total and abiotic immobilization. These data provide a rapid assessment of
the abiotic versus microbial sink strengths for $P_i$, but do not represent gross rates.

### 2.6 Calculations of gross rates of soil P dynamics

Calculation of gross IPD rates followed the mass balance equations of Kirkham and Bartholomew (1954), as later
applied by others for soil gross P fluxes (Kellogg et al., 2006;Mooshammer et al., 2012). However, in previous





work abiotic process contributions were not removed from the final data. To calculate gross $P_o$ mineralization, gross rates of $P_i$ desorption had to be corrected for in live soils. This was performed by using IPD calculations for influx (GI, gross influx; equation 1) for sterile soils (abiotic influx by $P_i$ desorption) and live soils (total $P_i$ influx), and taking the difference as biotic influx (i.e. gross $P_o$ mineralization). The same procedure was performed for tracer efflux (GE=gross efflux; equation 2) calculating gross immobilization fluxes for live soils (total $P_i$ efflux)

and sterile soils ($P_i$ sorption), the difference providing gross rates of microbial $P_i$ immobilization.

$$\text{Gross influx: } GI = \frac{C_{t2} - C_{t1}}{t_2 - t_1} \times \frac{ln\left(\frac{SA_{t1}}{SA_{t2}}\right)}{ln\left(\frac{C_{t2}}{C_{t1}}\right)} \qquad \text{(Eq. 1)}$$

$$\text{Gross efflux: } GE = \frac{C_{t1} - C_{t2}}{t_2 - t_1} \times \left(1 + \frac{ln\left(\frac{SA_{t2}}{SA_{t1}}\right)}{ln\left(\frac{C_{t2}}{C_{t1}}\right)}\right) \qquad \text{(Eq. 2)}$$

where $t_1$ and $t_2$ represent incubation time (4 and 24 h; in days), $C$ the soil $P_i$ concentration (in µg $P_i$ g$^{-1}$ soil dry weight), $SA$ the specific activity (in Bq µg$^{-1}$ $P_i$) and LN the natural logarithm. Gross rates are therefore in µg $P_i$ g$^{-1}$

$^{1}$ soil dry weight d$^{-1}$. Due to the relatively rapid decline in $^{33}$P activity by radioactive decay, all data were decay corrected back to the start of each experiment, i.e. the time point of tracer addition to the soil. This was done according to equation 3.

$$N_{t0} = \frac{N_t}{e^{-\lambda t}} \qquad \text{(Eq. 3)}$$

where $N_{t0}$ is the decay corrected $^{33}$P activity in a sample (in Bq), $N_t$ the measured $^{33}$P activity at time of liquid

scintillation counting, $t$ is time (in days) elapsed between tracer addition and $^{33}$P activity measurement, $e$=2.71828 and λ the decay constant of $^{33}$P (0.0273539).

## 2.7    Statistics

Regressions were performed in Sigmaplot 13.0 (Systat Software, Inc.) and group differences were tested by one-

way and two-way ANOVA followed by Tukey's HSD test in Statgraphics Centurion XVIII (Statpoint Technologies, Inc.). Variance homogeneity was tested by Levene's test and if necessary data were log, square root or rank transformed to meet assumptions of homoscedasticity and normal distribution.

## 3    Results

### 3.1    Soil characterization

Temperate grassland soils had a pH between 6.3 and 6.8, with a silt loam to sandy loam texture (Table 2). Soil organic C contents ranged between 48 and 127 mg C g$^{-1}$, soil N from 2.3 to 5.0 mg N g$^{-1}$ and soil total P from 0.44 to 0.82 mg P g$^{-1}$. Tropical forest soils had a pH between 4.1 and 4.2, and soil texture varied between silt, silt loam and sandy loam. Soil organic C contents were lower, at 26 to 31 mg C g$^{-1}$, soil N ranged from 2.2 to 2.6 mg N g$^{-1}$,

and soil total P from 0.09 to 0.17 mg P g$^{-1}$. Organic P comprised a larger fraction of total P in tropical forest soils (64-76%) than in temperate grassland soils (22-57%). Extractable soil $P_i$ was higher in temperate grasslands (4.2-13.1 µg P g$^{-1}$ soil dry weight) compared to tropical forest soils (0.07-0.13 µg P g$^{-1}$ soil dry weight). Acid phosphomonoesterase activities of tropical forest soils (1396-2346 nmol MUF released g$^{-1}$ dry weight h$^{-1}$) markedly exceeded those in temperate grasslands (233-256 nmol MUF released g$^{-1}$ dry weight h$^{-1}$).






### 3.2    Abiotic controls: soil sterilization efficiency

A separation of biotic and abiotic processes is based on the comparison of gross rates using the IPD assay in live versus autoclaved soils, where the latter should not exhibit any microbial activity (no FDA hydrolysis activity) and no extracellular enzyme activities (no MUF-phosphatase activity), in order to serve as abiotic controls. An

incomplete inhibition of extracellular phosphatase activities would lead to an underestimation of biological processes and therefore of gross $P_o$ mineralization. Our results show that two consecutive treatments of the soils by autoclaving, with a 24 hours incubation in between, effectively reduced microbial metabolic activity as shown by the reduction in soil FDA hydrolysis by 90% in soil 4 and by 97-99% in all other soils (Fig. 2). Autoclaved soils did not show any increase in soil microbial activity during the two days of incubation. On the contrary, the

inhibition of FDA hydrolysis even increased from 1 hour (all soil average: 94%) towards 24 and 48 hours after sterilization (average: 97-99%). The inhibition of extracellular acid phosphatase activity was almost complete in tropical soils (95-97%) and strongly reduced in temperate soils (79-80%). Similar to FDA hydrolysis the extent of inhibition of phosphatase activity increased from day 0 (average: 86%) to day 1 and 2 (average: 88-89%, Fig. 2). Autoclaving enhanced available $P_i$ by 1.86 ±0.32-fold (mean±1SD) in temperate soils and by 1.65±0.36-fold in

the tropical soils (Fig. S1). Sterilized soils can therefore serve as abiotic process controls, where microbial activity does not contribute to $P_i$ immobilization, and $P_o$ mineralization was largely halted as shown by strongly decreased phosphatase activities.

### 3.3    Comparison of isobutanol fractionation and direct measurements of $P_i$ and [33]P activity

Soil $P_i$ concentrations measured by the malachite green method directly in acidified bicarbonate extracts were compared to those measured after isobutanol fractionation by phosphomolybdate blue reaction. Both approaches yielded similar soil $P_i$ concentrations, and the relationship showed no bias (slope = 0.979±0.033, mean±1SE), with a coefficient of determination of 0.92 (Fig. 3B). The malachite green method is much more sensitive and therefore produced more reliable results for the low-P soils from the three tropical forests. Moreover, the relationship

between [33]P recoveries by isobutanol fractionation and by direct measurements in acidified bicarbonate extracts had a slope less than 1 (slope=0.875±0.010; Fig. 3A), indicating no significant formation of [33]$P_o$ during soil incubations. We also found no [33]$P_o$ formation in other soils using the same measurement protocols, e.g. from the Jena biodiversity experiment (82 plots of temperate grassland varying in soil texture and plant biodiversity, slope=0.891±0.017) and from French Guyana (24 soils from two primary forest regions, with soils sampled across

topographic gradients, slope=1.043±0.020) (same regression types as in Fig. 3A; data not shown). The specific activities of $P_i$ were indistinguishable between both approaches for temperate soils (slope=0.977±0.064, $R^2 = 0.93$, P<0.0001; Fig. 3C) but varied strongly for the tropical soils, where soil $P_i$ measurements in the isobutanol fraction were at or below the limit of detection of the phosphomolybdate blue method.

### 3.4    Sensitivity of the IPD assay

The sensitivity of this assay is greatly improved relative to traditional ones, by using a combination of bicarbonate extractions and malachite green $P_i$ measurements. The detection limit of the IPD approach was 0.12 µg P g$^{-1}$ soil dw d$^{-1}$, based on three times the standard deviation of gross $P_o$ mineralization measured for the three tropical soils, and therefore fully suitable across all soil types tested so far. However, the precision suffers from IPD equations

that combine uncertainties from four measurements, two $P_i$ concentrations and two radioactivity measurements for the two time points in live as well sterile soils. The coefficients of variation (CV) ranged between 1.0 and 22.1%



(average 10.0%) for $P_i$ concentration across temperate and tropical soils, and between 1.5 and 22.1% (average 9.6%) for SA, the two major input variables into the IPD equation. CVs increased towards lower $P_i$ concentrations and higher SA values, i.e. closer to the detection limit of the malachite green method. The CVs might be reduced

by working with larger soil aliquots (increase from 2 to 5 or 10 g soil fresh weight) and by duplicate measurements of all samples. Purely methodological CVs were lower, at about 2.5 and 0.9% for $P_i$ measurements by malachite green in the range 3-12 and 12-120 μM, respectively, and 0.8% for liquid scintillation counting. Therefore, much of the shown variability derived from differences between biological soil replicates. However, the variability found here compares well with CVs published for soil $P_i$ concentrations of 2-10% (Bünemann et al., 2007) and 20-25%

(Bünemann et al., 2012), and CVs for measured E values that are calculated from $P_i$ concentrations and $^{33}P$ recoveries analogous to SA values ranged between 6-16% (Bünemann et al., 2007), 8-19% (Bünemann et al., 2012) and 9-10% (Randriamanantsoa et al., 2015) across a range of cultivated and non-cultivated soils from temperate to tropical regions. These variations naturally propagate into higher errors in the measured rates of soil P cycling and increase the limit of detection and the limit of quantification of the various methods.


### 3.5    Time kinetics

During the first hour of the incubation, we found a rapid drop in $^{33}P$ recovery and in the SA of $P_i$ (Fig. 4), while soil $P_i$ concentrations increased slightly (Fig. S1). Thereafter a dynamic equilibrium between added $^{33}P$ tracer and the soil $P_i$ pool was reached and concentrations of extractable $P_i$ remained constant. A plot of ln($^{33}P$ recovery)

versus time of both live and sterile soils showed that the consumption of $^{33}P$ occurred linearly between 4 and 48 h in the temperate soil and between 2 and 24 h in the tropical soil. Similarly, the plot of ln(SA of $P_i$) versus time showed a linear relationship from 4 to 48 h in the temperate soil and for 2 to 48 h in the tropical soil, particularly in live soils (Fig. 4), showing constant dilution of the isotopic signature of the pool over time. The regressions became insignificant in the sterile tropical soil, as $^{33}P$ recovery and SA declined more slowly. The data clearly

show that abiotic $^{33}P$ processes (i.e. decreases in $^{33}P$ recovery and SA of $P_i$ over time in sterile soils) occurred, particularly in the temperate soil, and this over a prolonged period of time. More importantly, the dynamics of abiotic $^{33}P$ processes changed over time: rapid abiotic immobilization during the initial 0-4 h was followed by a period of slower but linear tracer immobilization.

### 3.6    $^{33}P$ pool dilution rates of abiotic and biotic processes

We calculated gross $P_i$ influx and efflux rates for live and sterile soils. Calculated rates of sterile soils provide estimates of gross rates of soil $P_i$ sorption and desorption, and the difference between live and sterile soils give the biotic influx (gross $P_o$ mineralization) and efflux (gross microbial $P_i$ uptake). Gross $P_o$ mineralization significantly differed between soils, with two out of three temperate soils (0.48 to 2.03 μg P $g^{-1}$ dw $d^{-1}$) exhibiting higher rates

than two out of three tropical soils (0.08 to 0.15 μg P $g^{-1}$ dw $d^{-1}$). Gross rates of $P_i$ sorption in temperate soils (2.06 to 6.14 μg P $g^{-1}$ dw $d^{-1}$) were higher than in tropical soils (0.15 to 0.32 μg P $g^{-1}$ dw $d^{-1}$), and a similar trend was found for gross rates of microbial $P_i$ uptake (temperate: 0.44 to 1.13 μg P $g^{-1}$ dw $d^{-1}$, tropical: 0.05 to 0.12 μg P $g^{-1}$ dw $d^{-1}$; Fig. 6B). Gross rates of soil $P_i$ desorption were significantly higher in temperate soils (1.44 to 3.63 μg P $g^{-1}$ dw $d^{-1}$) than in tropical soils (0.04-0.14 μg P $g^{-1}$ dw $d^{-1}$, Fig. 6A). The relative contribution of $P_o$ mineralization to

total $P_i$ release into the soil $P_i$ pool ranged between 25.0 and 73.8%, with two tropical P-poor soils showing the highest contributions (Fig. 6C). Contributions of biological processes to gross $P_i$ immobilization did not differ between soils (range 11.5% to 34.9%).



### 3.7 Net 33P immobilization by abiotic and biotic processes

Abiotic net $^{33}$P immobilization (net soil P fixation) increased markedly from 0 to 48 h in the grassland soil (17 to 58% of added tracer), while it reached 83% almost instantaneously in tropical soil and further increased to 90% after 48 h (Fig. 5A). Similar patterns were found across all 6 soils, with significantly higher abiotic net immobilization in tropical than temperate soils, increasing in both with time from 0 to 4 and 24 h (Fig. 5C). Biotic (microbial) net $^{33}$P immobilization ranged from 3 to 8% in the tropical soil and 8 to 17% in the temperate soil in

the time kinetics experiment, with a significant increase in the temperate but not in the tropical soil (Fig. 5B). Similarly, biotic net $^{33}$P immobilization was low but increased with time in all three tropical soils (3 to 6%), while it was significantly higher in temperate soils but increased (soil 6) or decreased (soil 2 and 4) with time (Fig. 5D). Microbial immobilization was very fast, with almost instantaneous $^{33}$P uptake by microbes (sampling at 0 h), ranging between 3% (tropical soils) and 15-38% (temperate soils). Given the strong changes in both abiotic and

biotic net $^{33}$P immobilization, we suggest that it is best to measure them after 24 (up to 48) h.

Sequential extraction-liquid chloroform-extraction (sECE) allowed to directly follow net $^{33}$P uptake by microbes, whereas biotic net $^{33}$P immobilization was estimated indirectly as the difference in net $^{33}$P immobilization by live and sterile soils. In the two measured soils, sECE estimates of microbial net $^{33}$P uptake were higher than the microbial net $^{33}$P immobilization estimates (temperate soil: 24.6% vs. 16.0%, and tropical

soil: 16.8% vs. 7.5%, for direct and indirect estimates, respectively).

### 3.8 Physicochemical and biological controls on soil $P_i$ processes

Gross $P_o$ mineralization was strongly positively correlated with total soil P ($R^2$=0.87, P<0.01, Fig. 7A) and to total as well as extractable soil $P_i$ concentration ($R^2$>0.83, P<0.05, Fig. 7B) but not to soil organic P or its contribution

to soil total P, nor to soil organic C, soil texture or soil acid phosphatase activity (Table S1). Gross abiotic $P_i$ release rates through desorption and dissolution were strongly positively related to total soil P and bicarbonate soil $P_i$ ($R^2$=0.97 and 0.98, respectively, both P<0.001, Fig. 7C and Table S1), but not to other parameters such as soil pH, soil texture, and soil organic C content. Gross $P_i$ sorption rates exceeded gross $P_i$ desorption rates approximately 2-fold, but both were strongly related ($R^2$=0.99, P<0.001, Fig. 7E). Gross $P_i$ sorption rates were strongly positively

related to soil total P ($R^2$=0.96, P<0.001, Fig. 7D), soil total $P_i$ ($R^2$=0.88, P<0.05, Table S1) and bicarbonate soil $P_i$ ($R^2$=0.99, P<0.001, Table S1), but neither to soil pH, soil organic C, nor to clay content or soil texture. Abiotic net $P_i$ immobilization was most strongly and negatively related to soil pH ($R^2$=0.95, P<0.001, Fig. 7L) and weakly to soil $P_i$ sorption ($R^2$=0.59, P=0.073, Fig. 7J). Gross microbial $P_i$ uptake rates were directly proportional to microbial biomass P measured by sECE ($R^2$=0.95, P<0.01, Fig. 7G), and positively related to net microbial $P_i$

immobilization ($R^2$=0.85, P<0.01, Fig. 7I). We found a negative curvilinear relationship between net immobilization rates by sorption and microbes ($R^2$=0.97, P<0.001, Fig. 7F).

### 4 Discussion

About a decade ago Kellogg et al. (2006) compared two IEX/ID techniques with an IPD approach, identifying

several biases of the different approaches and making recommendations for further development. The authors highlighted IPD approaches with soil extractions using 0.5 M sodium bicarbonate as best suited, for potentially any type of soil. However, this approach is currently underused and had issues with abiotic controls. IPD methods





are state-of-the-art to measure gross processes of soil N cycling (Murphy et al., 2003), but have rarely been applied to soil P cycling processes (Mooshammer et al., 2012;Di et al., 2000;Kellogg et al., 2006). We here present a novel

and versatile approach to derive quantitative estimates of highly important soil P cycling processes that drive soil P availability in low to high P soils. The approach quantifies abiotic net soil $P_i$ fixation and net soil microbial $P_i$ immobilization, as well as gross rates of soil $P_o$ mineralization and the abiotic release of $P_i$ from non-extractable soil $P_i$ pools ($P_i$ desorption and dissolution), both causing gross influx of $P_i$ into the soil available $P_i$ pool. Furthermore, gross rates of $P_i$ immobilization by soil sorption and precipitation and by microbial uptake processes

are derived from the same data by calculating the efflux from the soil $P_i$ pool in sterile soils (abiotic) and in live minus sterile soils (biotic processes), respectively.

In contrast to many earlier IEX/ID assays the IPD approach presented here is based on real isotope pool dilution theory (Kirkham and Bartholomew, 1954), and not on curvilinear extrapolation of E values (Table 1). Moreover, IEX/ID assays of $P_o$ mineralization necessitate steady-state conditions (constant $P_i$ and microbial

biomass P pools, and constant rates of isotope exchange and respiration) to allow extrapolation of short-term exchange processes to the full length of the incubation experiments, whereas IPD approaches can accommodate non-steady state conditions as caused by flush effects and disturbances (Mooshammer et al., 2017) or as induced by addition of organic matter. The equations to estimate IPD rates can easily be solved for soils where target pool concentrations increase (net mineralization) or decrease (net immobilization) over time and where microbial

biomass P changes (Kirkham and Bartholomew, 1954), and do not necessitate constant pool sizes as wrongly suggested previously (Bünemann, 2015;Randhawa et al., 2005).

### 4.1   Soil sterilization

$^{33}$P IPD experiments in soils differ from the more common $^{15}$N IPD variants for gross N processes (Murphy et al.,

2003), since the persistence of abiotic P processes over time (Figs. 4 and 5) needs to be accounted for via the use of sterile soils. Our data clearly show that the dynamics of abiotic $^{33}$P processes change over time. Therefore, the IPD rates in the sterile soils need to be measured over the same time period and under similar environmental conditions as in the live soils. It is insufficient and erroneous to extrapolate from short-term (100 min) incubations run under very different conditions to correct for abiotic processes in the respective live soil incubations

(suspension versus moist soil assays). Bünemann et al. (2007) clearly demonstrated that batch incubations (1:10 (w:v) soil: water suspensions) have higher water-soluble and isotopically exchangeable $P_i$ concentrations (measured as extractable Pi and as E values) and tended to have higher tracer recoveries (measured as r/R, i.e. water-soluble $^{33}P_i$ recovered relative to total $^{33}P_i$ added) compared to moist soil incubations.

We chose autoclaving as a sterilization procedure as other procedures only reduce or eliminate microbial

activity (gamma irradiation, azide, mercuric chloride, toluene or chloroform treatment) but do not stop extracellular enzyme activities (Blankinship et al., 2014;Wolf et al., 1989;Tiwari et al., 1988;Oehl et al., 2001b). Given that $P_o$ mineralization is mediated by extracellular phosphatases, previous isotope experiments using short-term batch experiments with or without inhibitors or γ-irradiation therefore did not allow to separate abiotic and biotic mechanisms of $P_i$ release in soils. Some treatments such as chloroform fumigation lyse microbial cells releasing

intracellular phosphatases into the soil environment, potentially stimulating $P_o$ mineralization in abiotic controls, although increases in phosphatase activity have rarely been documented in fumigated soils (Blankinship et al., 2014;Klose and Tabatabai, 2002;Tiwari et al., 1988). While application of phosphatase inhibitors might be another viable option, we are only aware of one study testing this; application of silver nanoparticles to soils showed a





general inhibitory effect on soil enzymes (Shin et al., 2012). Previous tests in our laboratory with two commercial

phosphatase inhibitor cocktails (Sigma-Aldrich) at 10-fold of the recommended final concentration did not significantly decrease IPD rates in two soils (data not shown), indicating an insufficient inhibition of extracellular phosphatases. In contrast, autoclaving soils twice was highly efficient in suppressing biological activities, and those soils had no or very low extracellular enzyme activity and no residual microbial metabolic activity. Previous studies showed (almost) total inhibition of hydrolytic enzyme activities (including phosphomonoesterases) by

autoclaving, in a wide range of arable, grassland and forest soils (Serrasolsas et al., 2008;Kedi et al., 2013;Blankinship et al., 2014;Tiwari et al., 1988). Other studies demonstrated successful killing of bacterial and fungal cells in soils by autoclaving (Carter et al., 2007;Blankinship et al., 2014;Serrasolsas and Khanna, 1995b;Alphei and Scheu, 1993)). Only in one study phosphodiesterase activity was resistant to autoclaving in one smectitic soil (Carter et al., 2007). Such resistance could either result from strong stabilization of soil

phosphodiesterases by clay-organic matter-complexation in this smectitic soil type or from the known heat stability of some nucleases which belong to the group of phosphodiesterases. However, the final step in $P_o$ mineralization is catalyzed by phosphomonoesterases, which were inactivated by autoclaving in all soils tested so far.

It must be noted that autoclaving could potentially alter the physicochemical properties of soils, thereby affecting abiotic sorption-desorption kinetics. Despite this, in previous studies autoclaving up to two times and

steam sterilization did neither affect the cation exchange capacity, nor base saturation, soil surface area, contents of total organic carbon and total nitrogen, and only slightly soil pH (Wolf et al., 1989;Tanaka et al., 2003;Serrasolsas and Khanna, 1995b). In our study autoclaving caused a pulse of labile P into the available soil P pool due to the lysis of microbial biomass (Fig. S1), as has also been demonstrated for P and N by Serrasolsas and Khanna (1995a, b). Soil $P_i$ concentrations increased significantly in the autoclaved soils studied here, but only by

an average of 1.86-fold in the three temperate soils and by 1.65-fold in the three tropical forest soils, which was in the range found by others, e.g. 1.3- to 1.6-fold (Skipper and Westermann, 1973) and 1.5- to 1.6-fold (Anderson and Magdoff, 2005) but lower than reported elsewhere, e.g. 2.6- to 11-fold (Serrasolsas and Khanna, 1995a). In a study, where biotic and abiotic sinks (sorption, microbial uptake) and sources (desorption, microbial lysis) were not partitioned, increased P desorption and decreased soil P sorption capacity after autoclaving could largely be

explained by the release of $P_i$ due to microbial lysis and the concurrent loss of microbial P sink capacity (Serrasolsas et al., 2008). Autoclaving was also demonstrated to increase the tracer recovery (r/R) and decrease the velocity of its decline over time as expected due to loss of microbial biomass. Autoclaving therefore slightly affects the soil $P_i$ pool, but most likely has no or minor effects on its abiotic sorption/desorption dynamics while it inhibits biological reactions. Moreover, natural fluctuations in soil $P_i$ concentrations can be large, e.g. 2-fold

between field replicates (Bünemann et al., 2012) and >40-fold for diffusive phosphate fluxes in soil replicates determined by microdialysis (Demand et al., 2017), and are therefore similar or substantially higher than the effects observed due to autoclaving. Finally, as stated earlier, changes in $P_i$ concentration caused by autoclaving can easily be accounted for in IPD approaches, as long as abiotic process rates remain unaffected. Autoclaving therefore allows to correct live soil IPD rates for abiotic process contributions, and thereby to calculate gross rates of $P_o$

mineralization and microbial $P_i$ uptake.

### 4.2 Soil $P_i$ extraction using bicarbonate

Similar to $^{15}N$ IPD assays, where salt extractions are employed to target the available inorganic or organic N pool
(Murphy et al., 2003;Wanek et al., 2010;Hu et al., 2017), we focused on the potentially bio-available, salt-





extractable $P_i$ pool that reflects the plant- (and microbial) accessible amount of soil $P_i$ better (Fardeau et al., 1988;Olsen et al., 1954;Horta and Torrent, 2007) than the water extractable $P_i$ pool that is mostly assessed with soil IEX/ID methods. The applied 0.5 M NaHCO$_3$ extraction (pH 8.5, Olsen P) promotes the displacement of $P_i$ (and the extraction of labile $P_o$), particularly from Al-Fe (hydr)oxides and soil organic matter, by competition of bicarbonate anions with $P_i$. The underlying process is an increase of the negative charge on surfaces and a decrease

of the concentration and activity of $Ca^{2+}$ and $Al^{3+}$, thereby increasing P solubility in acid to alkaline soils (Horta and Torrent, 2007;Schoenau and O'Halloran, 2008;Demaria et al., 2005). Several studies compared soil P tests like Bray III, resin P, and Olsen-P to soil water $P_i$ and plant P uptake in order to assess how well they reflect the available $P_i$ pool. These studies demonstrated that soil tests like bicarbonate extractions (Olsen-P), resin P and DGT (diffusive gradients in thin films technique) closely resembled the SA values of $P_i$ extracted by water or 10

mM CaSO$_4$ or from plants (Six et al., 2012;Fardeau et al., 1988;Demaria et al., 2005). Others further showed that isotopically exchangeable $P_i$ in soil water extracts (E values) and those extracted by plant roots in plant growth experiments (L values) also were strongly related (Bühler et al., 2003;Frossard et al., 1994). Bicarbonate extracted 8- to 22-fold greater amounts of exchangeable $P_i$ compared to water and SA of $P_i$ in bicarbonate extracts reached 66-90% of the SA values measured in soil water extracts (Demaria et al., 2005). IPD approaches require fast

extractions to quickly terminate the assay after 4 and 24 h, which renders water extractions (generally 16 h), resin P (16 h) and DGT (up to 48 h in low P soils; (Six et al., 2012)) impossible. Bicarbonate extractions only take 30-60 min and therefore represent a viable alternative. Moreover, it makes the IPD assay on average 8-fold more sensitive as a greater amount of exchangeable $P_i$ is extracted by bicarbonate than with water (Kleinman et al., 2001).

In some studies, the $^{33}$P activity in microbial biomass is included with water extractable $P_i$ as a kind of labile pool in the calculations of isotope exchange kinetics, although SA dynamics in water-extractable $P_i$ and in microbial P often did not converge (Walbridge and Vitousek, 1987;Bünemann et al., 2007;Kellogg et al., 2006;Achat et al., 2010). Microbial P here is not considered to be bio-available during the short-term incubations over 24 or 48 h as it can only engage in IPD if microbes die and turnover. This only happens to a significant extent

with incubation periods greater than a few days or weeks (Oehl et al., 2001a;Kouno et al., 2002), since soil microbial biomass turnover times are ranging between 30 to 300 days (Spohn et al., 2016a;Spohn et al., 2016b).

### 4.3   Microbial P dynamics

We observed very fast microbial $P_i$ immobilization in live soils (within minutes; extraction started directly after

tracer addition), causing net immobilization of $^{33}$P by 3-38%. Similar results were reported within 1.5 to 4 h by others, ranging from 6-37% (Bünemann et al., 2012;Kellogg et al., 2006). This has two major repercussions: (i) rapid uptake might cause microbial $P_i$ assimilation and efflux or exudation of $^{33}$P$_o$ metabolites without microbial death and turnover. However, the comparison between specific activities and $^{33}$P recoveries of the direct measurement and after isobutanol fractionation (see below, and Fig. 3) showed that no significant release of

microbial $^{33}$P$_o$ occurred during the 24 and 48 h incubations. The short extraction times used in this study also decrease the likelihood of significant hydrolysis of $P_o$ compounds. (ii) Rapid microbial $^{33}$P$_i$ uptake clearly rules out the use of $P_o$ mineralization assays that measure abiotic IEX/ID in short-term batch experiments (100 min) without addition of a microbicide or without prior sterilization and then extrapolate these "abiotic" process rates to the full experimental duration.



Microbial $P_i$ uptake can be derived indirectly as the difference in $^{33}P$ recovery between live and sterile soils (Fig. 5, this study), more directly by sECE (this study), by parallel water or bicarbonate extraction with and without addition of liquid chloroform or hexane (measuring resin strip or extractable $P_i$), or by chloroform fumigation extraction (Bünemann et al., 2012;Oberson et al., 2001;Oehl et al., 2001a;Spohn and Kuzyakov, 2013). Microbial net $^{33}P$ immobilization measured by direct sECE was higher relative to the difference in $^{33}P$ immobilized

in live minus sterile soils, pointing towards (i) overestimation of microbial net $^{33}P$ immobilization by sECE due to incomplete extraction of non-microbial $^{33}P_i$ by one-time bicarbonate extraction prior to sECE, or (ii) overestimation of abiotic sorption processes by autoclaving. In favor of (i) repeated extractions of soils with Bray I-extractant showed that soils continued to release P at lower rates in subsequent extractions after readily extractable P was removed by the first extraction (Serrasolsas et al., 2008;Messiga et al., 2014). Repeated

extractions with bicarbonate also showed that the first extraction only removed 67-78% of the $^{33}P_i$ that was extractable with three consecutive extractions (D. Wasner, data not shown). In favor of (ii) (Kellogg et al., 2006) found higher net $^{33}P$ immobilization or sorption in sterile compared to live soils. This was interpreted as a lack of microbial competition for P in sterile soils. However, we found a weak positive relationship (R=0.749, P=0.087; Table S1) between gross microbial $P_i$ uptake and gross $P_i$ sorption. This opposes the idea of strong competition

between sorption and microbial uptake on the basis of gross process measurements. Another possible mechanism underlying (ii) could be changes in soil structure and reactive surfaces enhancing soil P sorption. Delineation of the causes could be performed by a comparison of sECE with parallel assessments of microbial $^{33}P$ uptake, using a comparison of $^{33}P$ in bicarbonate versus bicarbonate+liquid chloroform or bicarbonate+liquid hexane extracts. Given the continued extraction of $P_i$ from exchangeable $P_i$ pools in serial extraction tests, parallel determination

of microbial P and $^{33}P$ seems favorable relative to sequential extractions for microbial P determination.

### 4.4     Comparison of isobutanol fractionation with direct measurements of $P_i$ and $^{33}P$ activity

We showed that $^{33}P$ IPD assays can be performed specifically on the $P_i$ pool using isobutanol fractionation in high P soils. However, due to low production or persistence of $^{33}P_o$, results closely conformed with measurements run

without $P_i$-$P_o$ fractionation by malachite green and direct $^{33}P_{total}$ estimates. This was ascertained for forest soils from French Guyana and Costa Rica, and for grassland soils from Austria and Germany (data not shown for French Guyana and Germany). Isobutanol fractionation has previously been applied in radiotracer studies on P dynamics in soils (Kellogg et al., 2006) and litter (Mooshammer et al., 2012), to ascertain the separation of $P_i$ from any possible radiolabeled $P_o$ contaminant, however without comparison to SA in unfractionated bicarbonate extracts.

Oehl et al. (2001a) also applied isobutanol fractionation to water extracts of fumigated and control soils, demonstrating that with long extraction times (16 h), $^{33}P_i$ activities in water extracts with and without isobutanol fractionation were comparable. It was suggested that $^{33}P_o$ possibly released during fumigation was cleaved by soil phosphatases during extraction. This may not apply for short-term extractions (e.g. 0.5 M NaHCO$_3$ for 30 min, as used in this study) where hydrolysis by phosphatases would not necessarily occur due to short contact times.

Measurements of $^{33}P$ isotope pool dilution in soils based on bicarbonate extracts can therefore be interchangeably be performed by (i) direct measurements of $^{33}P_{tot}$ and $P_i$ in acidified bicarbonate extracts and after (ii) isobutanol fractionation on $^{33}P_i$ and $P_i$. However, this needs to be validated for other types of soil, and may change significantly after longer incubation periods (weeks), when microbial $^{33}P_i$ uptake, assimilation and turnover causes the release of $^{33}P_o$ into the soil. The short cut by performing direct measurements of $P_i$ concentration and $^{33}P$ in

acidified bicarbonate extracts comes along with 4- to 10-fold greater sensitivity of the malachite green assay



relative to phosphomolybdate blue measurements of soil $P_i$. Another option to increase the measurement sensitivity for $P_i$ (and possibly also for $^{33}P_i$) for strongly sorbing low-P soils has been adopted by Randriamanantsoa et al. (2013), based on concentration of the phosphomolybdate blue complex from a large volume of extract into a smaller volume of hexane, with subsequent phase separation (Murphy and Riley, 1962). This allowed to decrease

limits of quantification of $P_i$ by 66-fold compared to the classical Murphy-Riley protocol, and 14-fold compared to the malachite green procedure (Randriamanantsoa et al., 2013) but involves the handling of large volumes of radiolabeled extracts.

### 4.5    Time kinetics

During the first minutes, equilibration between tracer and tracee was not achieved, indicated by the enhanced extractability of added tracer ($^{33}P_i$) relative to more strongly bonded native tracee (soil exchangeable $P_i$). The fast process of equilibration caused very rapid declines in SA of $P_i$ during the first few minutes. Thereafter, microbial uptake and soil P fixation caused a rapid draw down of extractable $^{33}P_i$ and thereby a further decrease in the SA of soil $P_i$ while soil $P_i$ concentrations did not change after the initial phase of tracer-tracee equilibrium (Fig. 4).

These processes slowed down within the first 1-2 h but did not seize, and declines in $^{33}P$ recoveries and in the SA of $P_i$ occurred over the whole incubation period, in sterile as well as live soils. Thereafter time kinetics of IPD were relatively constant between 4 and 24 h for both, temperate and tropical soils, as shown by the linearity of the relationship in a plot of $\ln(SA$ of $P_i)$ versus time. This linear relationship is conceptually different from the plot of $\log(recovery, r/R)$ versus $\log(time)$ in short-term IEX/ID batch experiments, that provides the parameter "n",

i.e. the slope or the rate of decline in tracer recovery due to sorption over time (Bünemann, 2015). Based on constant IPD rates in the above-mentioned time interval we advise to run $^{33}P$ pool dilution experiments for an incubation period of 4 to 24 h. This time frame is well within the linear range, as it lies after the rapid abiotic equilibration, and is long enough to allow significant pool dilution to occur for sensitive measurements of organic P mineralization. Longer incubation times are not recommended due to the risk of $^{33}P_o$ release from dying

microbes, potentially causing a $^{33}P_i$ reflux through remineralization, violating a major assumption of IPD theory. Strong increases of abiotic or microbial immobilized $^{33}P_i$ refluxes over time would cause a plateau or even an inversion of the SA kinetics, and the likelihood of $^{33}P_i$ refluxes increases with increasing incubation time. Indeed, in several long-term IEX/ID experiments lasting 15-68 days plateaus or even increases of SA have been observed, whereas in short-term incubations not (Oehl et al., 2001a;Kellogg et al., 2006;Bünemann et al., 2007;Bünemann

et al., 2012;Randriamanantsoa et al., 2015;Achat et al., 2010). Three arguments point against $^{33}P$ reflux from immobilized P pools during our short-term incubations (24 h): (i) we did not observe a plateau in SA of $P_i$, (ii) soil $^{33}P_i$ fixation increased over time indicating that fixed P was not becoming available again during these short-term incubations but was rather transferred to more strongly bound $P_i$ pools, and (iii) no $^{33}P_o$ release from microbes was found, which indicates that $^{33}P_o$ metabolites were not exuded by microbes (see *Comparison of isobutanol*

*fractionation and direct measurements of $P_i$ and $^{33}P$ activity*).

### 4.6    Comparison of $P_o$ mineralization rates with published values

The detection limit of the IPD approach was 0.12 µg P g$^{-1}$ soil dw d$^{-1}$. In comparison, the detection limits for gross $P_o$ mineralization by the IEX/ID approach were 0.20 µg P g$^{-1}$ soil d$^{-1}$ by the modified protocol including hexane

concentration of phosphomolybdate blue for tropical soils (Randriamanantsoa et al., 2015) and 0.6-2.6 µg P g$^{-1}$ soil d$^{-1}$ by the traditional IEX/ID approach on temperate soils (Bünemann et al., 2007). Values of gross $P_o$




mineralization measured via IPD in this study ranged between 0.08-0.15 µg P g$^{-1}$ soil dw d$^{-1}$ in tropical forest soils and 0.48-2.03 µg P g$^{-1}$ soil dw d$^{-1}$ in temperate grassland soils and were therefore well in the range of those compiled for IEX/ID measurements by Bünemann (2015) for 14 different soils, including temperate arable,

grassland and forest soils (0.1-12.6 µg P g$^{-1}$ soil dw d$^{-1}$) and one tropical arable soil (0.8 µg P g$^{-1}$ soil dw d$^{-1}$). To date, highest gross $P_o$ mineralization rates were reported for decomposing beech litter, i.e. 22.5-86.3 µg P g$^{-1}$ soil dw d$^{-1}$ (Mooshammer et al., 2012). Kellogg et al. (2006) found that gross $P_o$ mineralization rates based on IPD approaches tended to be higher than those calculated by IEX/ID experiments on the same soils. However, this is most likely due to the fact that the authors did not correct for abiotic process contributions in their IPD approach

and followed steady-state assumptions in their calculations. A direct comparison of the present IPD and the IEX/ID approaches on the same soils might help to clarify how far the approaches really deviate in their gross $P_o$ mineralization rate estimations.

### 4.7 Physicochemical and biological controls on soil Pi processes

We found that gross $P_o$ mineralization was strongly positively correlated to total soil P but not to soil organic P, soil organic C, soil texture or soil acid phosphatase activity. This indicates that gross $P_o$ mineralization might rather be driven by total P or organic P availability than by soil enzyme activity, and that total soil $P_o$ does not well represent the $P_o$ fraction accessible to soil phosphatases. A few studies demonstrated positive correlations between gross $P_o$ mineralization and soil $P_o$ (Lopez-Hernandez et al., 1998) or litter $P_o$ (or its inverse C:P; (Mooshammer

et al., 2012)). However, Wyngaard et al. (2016) did not find this relationship of gross $P_o$ mineralization with total soil $P_o$ but with the $P_o$ content of the coarse soil fraction only, which points into a similar direction as our results. Moreover, $P_o$ mineralization might be controlled rather by soil phosphodiesterases targeting DNA, RNA, teichoic acids and phospholipids, than by phosphomonoesterases that are responsible for the final extracellular dephosphorylation of $P_o$. In contrast to our results, positive relationships were found between gross $P_o$

mineralization and phosphomonoesterase activities in two studies (Spohn et al., 2013;Oehl et al., 2004), however not across studies (Bünemann, 2015). A larger set of soils varying in soil pH, texture and mineralogy might therefore provide better insights into the controls of soil $P_o$ mineralization, such as effects by extracellular phosphatase activity (phosphomonoesterases and phosphodiesterases), and the availability, stabilization and accessibility of organic P in soils, among others. Moreover, high $P_i$ availability (i.e., bicarbonate $P_i$) strongly

suppressed phosphomonoesterase activity in soils, causing a negative correlation between the enzyme activity and extractable $P_i$, while extractable $P_i$ was positively related to gross $P_o$ mineralization, indicating that high-$P_i$ conditions suppressed phosphatase production but not $P_o$ mineralization across these soils, which was also found as a positive correlation between gross $P_o$ mineralization and water-extractable $P_i$ by others (Schneider et al., 2017).

The contribution of gross $P_o$ mineralization to total $P_i$ supply including $P_i$ desorption from exchangeable

$P_i$ pools and dissolution ranged between 25 and 74%, with a trend towards larger contributions in low-P tropical soils (35-74%) compared to temperate soils (25-51%). This clearly demonstrates that biological processes contribute importantly to the $P_i$ supply in soils, particularly in low-P soils, as also pointed out by (Bünemann, 2015). In low-P forest soils biological processes were shown to dominate over physicochemical processes, while in P-rich forest soils abiotic processes controlled gross $P_i$ supply rates (Bünemann et al., 2016). It was also found

that the contributions of microbial processes decreased with soil depth, where in deep soils diffusive fluxes (i.e. gross $P_i$ desorption) dominated the soil $P_i$ supply due to low total $P_o$ contents relative to total P (Achat et al., 2012;Achat et al., 2013).



Gross abiotic $P_i$ release rates through desorption and dissolution were strongly positively related to total soil P and bicarbonate $P_i$, but not to other parameters such as soil pH, soil texture, and soil organic C content. In

contrast to the weak effects of soil pH and texture on gross soil $P_i$ supply, soil mineralogy and particularly oxalate-extractable Fe and Al as proxy for Fe-Al (hydr)oxides play a major role in controlling abiotic dynamics of phosphate ions in soils, across the full range from acidic to alkaline soils (Achat et al., 2016). Fe-Al (hydr)oxides provide large positively-charged surface areas in weathered soils that are highly reactive to phosphate ions, more so than clay minerals such as kaolinite, illite and others (Hinsinger, 2001;Regelink et al., 2015). Soil mineralogy

might therefore provide further interesting insights into the controls of abiotic processes as demonstrated by (Achat et al., 2011;Achat et al., 2016), but can also affect $P_o$ mineralization through strong effects on the sorption strength of organic matter and of $P_o$ compounds. Moreover, the elsewhere reported positive relations of $P_i$ availability and $P_i$ desorption with soil organic C contents was explained by competitive sorption of $P_i$ and SOC or DOC to reactive surfaces such as positively charged metal (hydr)oxides (Regelink et al., 2015;Achat et al., 2016).

Gross $P_i$ sorption rates exceeded gross $P_i$ desorption rates approximately 2-fold but both were strongly related, indicating close and rapid cycling of available $P_i$ through sorption-desorption processes. The observed rates indicate that soils immobilized more $P_i$ then they mobilized by abiotic processes, causing an intermediate draw down of available $P_i$ pools. The strong positive relationship between gross $P_i$ sorption rates and soil total P, soil total $P_i$ and bicarbonate soil $P_i$, and the lack of relationship with soil pH, soil organic C, clay content and soil

texture highlights again that specific soil minerals, particularly metal (hydr)oxides and to a lesser extent clay minerals such kaolinite, factors not fully captured by soil pH and soil texture alone, are responsible for $P_i$ sorption in soils (Regelink et al., 2015). In IEX/ID experiments it was found that the rate of abiotic $P_i$ depletion from soil solution through sorption  was positively related to Al-Fe (hydr)oxide content and negatively to soil organic C divided by Al and Fe oxide content (Achat et al., 2016;Tran et al., 1988). Interestingly, gross $P_i$ sorption was

weakly negatively related to abiotic net $P_i$ immobilization. This is because abiotic net $P_i$ immobilization was high in tropical soils showing strong $^{33}P_i$ sorption, but due to very small contents of available $P_i$ in those soils, gross $P_i$ sorption fluxes were lower than those in temperate grassland soils. This illustrates the major differences between measurements of net and gross processes, with both providing complementary information on soil P cycling processes. The strong negative relation between abiotic net $P_i$ immobilization and soil pH indicates that strongly

weathered, acid tropical soils have a higher P sorption and fixation capacity than temperate soils.

Gross microbial $P_i$ uptake rates were directly proportional to microbial biomass P measured by sECE, and positively related to net microbial $P_i$ immobilization. We also found a strong competition between net immobilization rates by sorption and microbes. This shows that freshly added radiotracer or native $P_i$ released by desorption or $P_o$ mineralization is competitively partitioned between microbial interception and uptake relative to

abiotic sorption, with greater $P_i$ immobilization potentials through sorptive reactions (28-92%) than through biological sinks (5-37%) in the soils studied here. The importance of rapid net uptake of tracer by soil microbes has been demonstrated also by other studies, e.g. (Bünemann et al., 2012). However, the presented IPD approach for the first time allowed to estimate gross rates of microbial $P_i$ uptake in addition to net microbial $P_i$ immobilization. Gross rates of microbial uptake were calculated from the IPD approach, not necessitating the

application of any extraction factor to calculate microbial biomass P from chloroform-labile P ($k_{EP}$-factor), which becomes necessary when studying net $P_i$ uptake over prolonged time periods in tracer experiments and for correction of net $P_o$ mineralization rates (Bünemann, 2015;Bünemann et al., 2007).



### 4.8    Application and modeling

The combination of this IPD assay with advanced numerical modeling approaches, as applied by Müller and Bünemann (2014), might further enhance the precision of estimates of simultaneously occurring soil P cycle processes and thereby advance the knowledge of major controls of the transformations and fluxes of this important nutrient in terrestrial ecosystems. There is an ever-increasing need of high quality data on soil P processes, even more so to calibrate terrestrial biogeochemical models and incorporate nutrient controls on plant productivity in

global models. This IPD approach may provide highly important quantitative data to implement soil P cycling processes into global biogeochemical models. This will further enhance our current understanding of nutrient controls on the global terrestrial C cycle and improve our capabilities to predict future changes by increasing discrepancies in N and P inputs into the terrestrial biosphere.

**Data availability.** The data of the different experiments are freely available upon request from the corresponding author.

**Author contributions.** The project was conceived and supervised by WW. DZ, JP and DW performed the measurements and data evaluation. WW wrote the manuscript with contributions from all coauthors.


**Competing interests**. The authors declare that they have no conflict of interest.

**Acknowledgements.** We are indebted to the Isotope Laboratory managers for access and training (Virginie Canoine, Markus Schmid).




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





Table 1. Comparison of traditional isotope exchange (IEX; synonymous with isotope dilution, ID) experiments and the novel isotope pool dilution (IPD) approach to measure organic P mineralization.

| Factor and approach | Isotope exchange (IEX/ID) | Isotope pool dilution (IPD) |
|---|---|---|
| Tracer addition and incubation period | $^{33}P$, $^{32}P$;<br>Several time points across several days to weeks and months | $^{33}P$, ($^{32}P$);<br>Two time points at 4 and 24 hours |
| Measured P pool | Water-extractable $P_i$ (sometimes including microbial $P_i$ by hexane/anion exchange resin strip method) | Bicarbonate-extractable $P_i$ and $P_o$ (excluding microbial P, which contributes to the gross efflux of $P_i$ but microbial P can be measured by sECE directly) |
| Abiotic controls | Abiotic controls measured in batch experiment with live soil: 100 min $P_i$ exchange experiment in soil suspension 1:10 (soil: water), ±$HgCl_2$ or sodium azide; microbial contributions in short-term experiment often not accounted for | Duplicate autoclaving for abiotic controls to kill microbial biomass and extracellular enzymes; treatment of abiotic controls similar as live soils in terms of tracer addition, incubation period and extraction |
| Microbial processes in abiotic controls | Microbial biomass active in abiotic controls if no microbicide added, extracellular phosphatases fully active (causing organic P mineralization in abiotic controls) | Microbial biomass and phosphatases deactivated by autoclaving (no/almost no P mineralization occurring in abiotic controls) |
| Pre-incubation of soils to equilibrate to moisture and temperature | Yes (to constant respiration – equilibrium conditions necessary) | Yes (not necessary) |
| Change in soil structure and P availability | No (if no microbicide is added) | Potentially yes, as autoclaving might increase available P by death of microbial biomass and soil structure might change by autoclaving |
| Numerical solution for $P_o$ mineralization | Isotopically exchangeable P within t minutes ($E_{(t)}$) derived as the inverse of the relative specific activity of phosphate in soil solution (water extractable $P_i$) over time in live soils. $E'_{(t)}$ derived for abiotic controls extrapolated from 100 min to length of full experiment, graphical solution of corrected data following (Fardeau, 1993). Differences in $E'_{(t)}$ and $E_{(t)}$ estimate gross $P_o$ mineralization | Calculation of IPD influx rates based on mass/isotope balance equations derived by (Kirkham and Bartholomew, 1954) for tracer: tracee experiments. Gross $P_o$ mineralization calculated as difference of IPD influx rates of live soils minus abiotic controls |



Table 2. Soil characterization of three temperate grassland soils (soil 2, 4, and 6) and three tropical lowland forest soils (soil 3, 5, and 7).

| Parameter | Unit | Temperate soils | | | Tropical soils | | |
|---|---|---|---|---|---|---|---|
| | | 2 | 4 | 6 | 3 | 5 | 7 |
| Soil pH (10 mM CaCl$_2$) | | 6.30 | 6.25 | 6.80 | 4.15 | 4.15 | 4.10 |
| Clay | (%) | 16.8 | 14.1 | 2.76 | 4.12 | 19.6 | 26.2 |
| Silt | (%) | 59.2 | 24.4 | 40.6 | 88.0 | 72.8 | 70.1 |
| Sand | (%) | 24.0 | 61.4 | 56.6 | 7.92 | 7.61 | 3.74 |
| Total organic C | (mg g$^{-1}$ DW) | 48.3 | 126.7 | 60.3 | 26.4 | 30.8 | 28.5 |
| Total N | (mg g$^{-1}$ DW) | 3.35 | 5.03 | 2.32 | 2.17 | 2.57 | 2.27 |
| Total P (TP) | (mg g$^{-1}$ DW) | 0.82 | 0.44 | 0.51 | 0.14 | 0.17 | 0.09 |
| Total organic P (TP$_o$) | (mg g$^{-1}$ DW) | 0.40 | 0.25 | 0.11 | 0.09 | 0.13 | 0.07 |
| Soil P$_i$ | (µg g$^{-1}$ DW) | 15.1 | 4.23 | 5.59 | 0.56 | 0.49 | 0.37 |
| TP$_o$ of TP | (%) | 49.1 | 56.5 | 22.3 | 64.2 | 75.7 | 76.4 |
| Soil C:N | | 14.4 | 25.2 | 26.0 | 12.1 | 12.0 | 12.5 |
| Soil C:TP$_o$ | | 121 | 507 | 548 | 293 | 237 | 406 |
| Soil N:TP$_o$ | | 8.4 | 20.1 | 21.1 | 24.1 | 19.8 | 32.5 |
| Phosphatase | (nmol MUF g$^{-1}$ DW h$^{-1}$) | 256 | 316 | 233 | 1396 | 1698 | 2346 |




Figure 1. Schematic representation of (A) major fluxes of soil P processes controlling the availability of inorganic
P ($P_i$) in soils, and of (B) the isotope pool dilution principle showing influxes of unlabeled $P_i$ ($^{31}P$) into the available
$P_i$ pool labelled by a spike of $^{33}P_i$, and efflux of $P_i$ at the ratio of $^{33}P$:$^{31}P$ as present in the target pool. Biotic and
abiotic processes of influx and efflux are presented. This causes (C) a decline in the specific activity of $P_i$ i.e.
$^{33}P_i$:$^{31}P_i$ declines over time in sterile soils (abiotic processes only) and live soils (biotic plus abiotic processes),
allowing to derive biotic contributions to overall gross fluxes. $TP_i$…total soil $P_i$, $TP_o$…total organic P, $TP_i$ includes
occluded and fixed P as well as primary mineral P, $TP_o$ includes occluded Po in aggregates. Avail…available. $P_i$
desorption includes $P_i$ dissolution from minerals, and $P_i$ sorption includes $P_i$ precipitation.

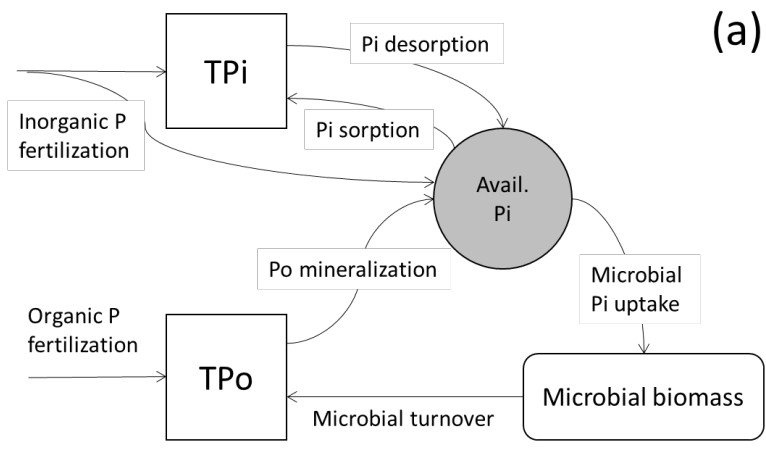

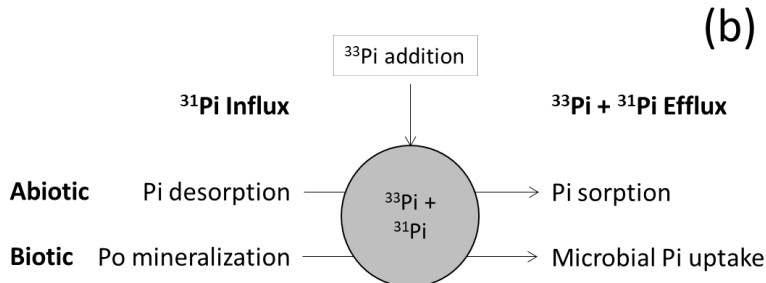

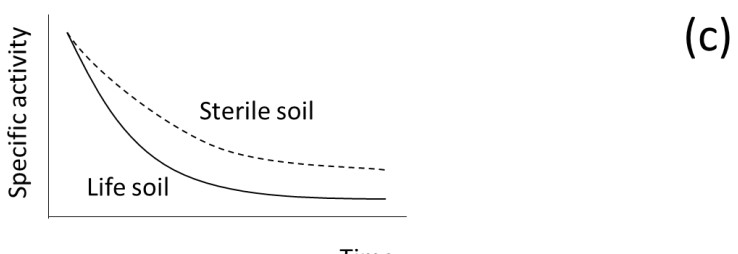





Figure 2. Response of soil enzyme activities to autoclaving: Percentage inhibition of (A) fluorescein diacetate
(FDA) hydrolysis as a proxy for the inhibition of live, cell-bound microbial enzyme activity and of (B) MUF-
phosphomonoesterase activity as a proxy for the inhibition of extracellular enzyme activity. Temperate grassland
soils (2, 4, 6) and tropical forest soils (3, 5, 7) were tested. Two-way ANOVA was calculated to test for the
factors soil, time (1, 24 and 48 hours after second autoclaving cycle, in open, grey and black bars, respectively)
and their interaction. P values are presented.

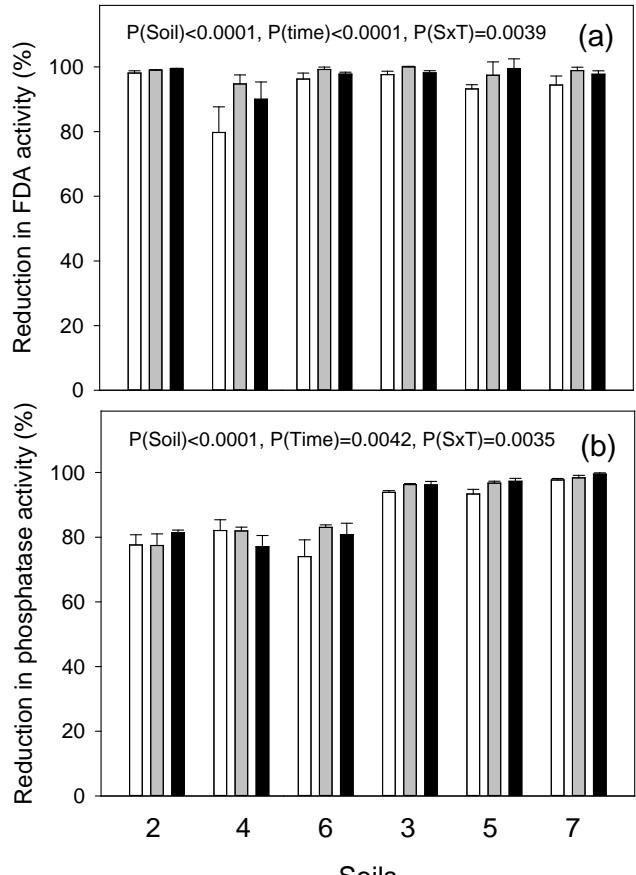



Figure 3. Relationship between (A) 33P recoveries as measured directly in acidified bicarbonate extracts and after
isobutanol fractionation, relative to the added tracer amount, and between (B) $P_i$ concentrations measured by the
malachite green method in acidified bicarbonate extracts and after isobutanol fractionation following the
phosphomolybdate blue approach. (C) Comparison of specific activities (SA) of $P_i$ measured in acidified
bicarbonate extracts and after isobutanol fractionation. Regression in (C) is only for temperate grassland soils
(closed circles) as for tropical forest soils (open circles) $P_i$ concentrations were close to the detection limit of the
phosphomolybdate method, impairing calculations of SA for isobutanol fractionation. Linear regressions are given
(slopes and intercepts ±1SD).

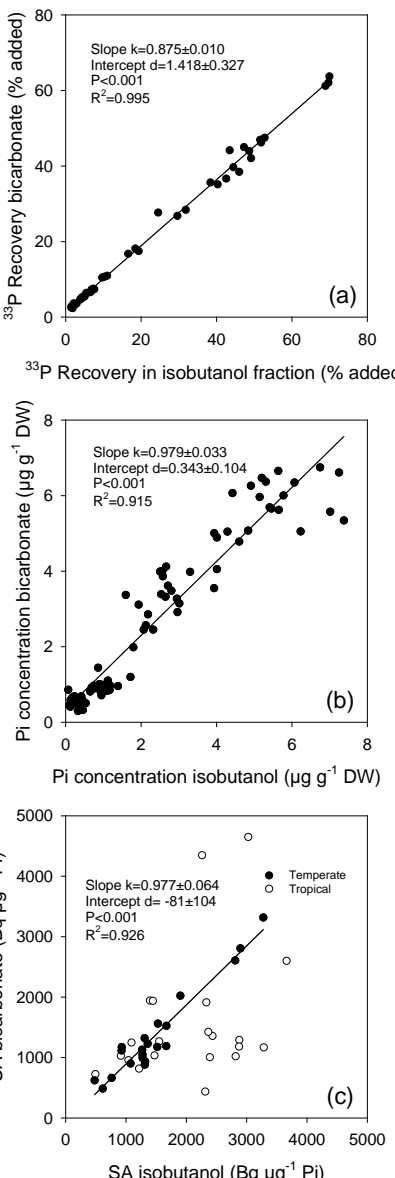





Figure 4. Test for linearity of change in $^{33}$P recoveries (A, B) and in specific activities of $P_i$ (C, D) over time, for a temperate grassland soil (A, C) and a tropical forest soil (B, D). Data presented are for $^{33}$P measured directly in

bicarbonate extracts of live soils (closed circles) and sterile soils (open circles), and are shown on y-axes in a logarithmic manner (LN). Regression lines follow exponential decay which in this linear – LN plot appears as straight line; dashed lines represent sterile soils and solid lines live soils. Regressions were calculated for the time interval 2 to 24 hours (tropical soil) and 4 to 48 hours (temperate soil).

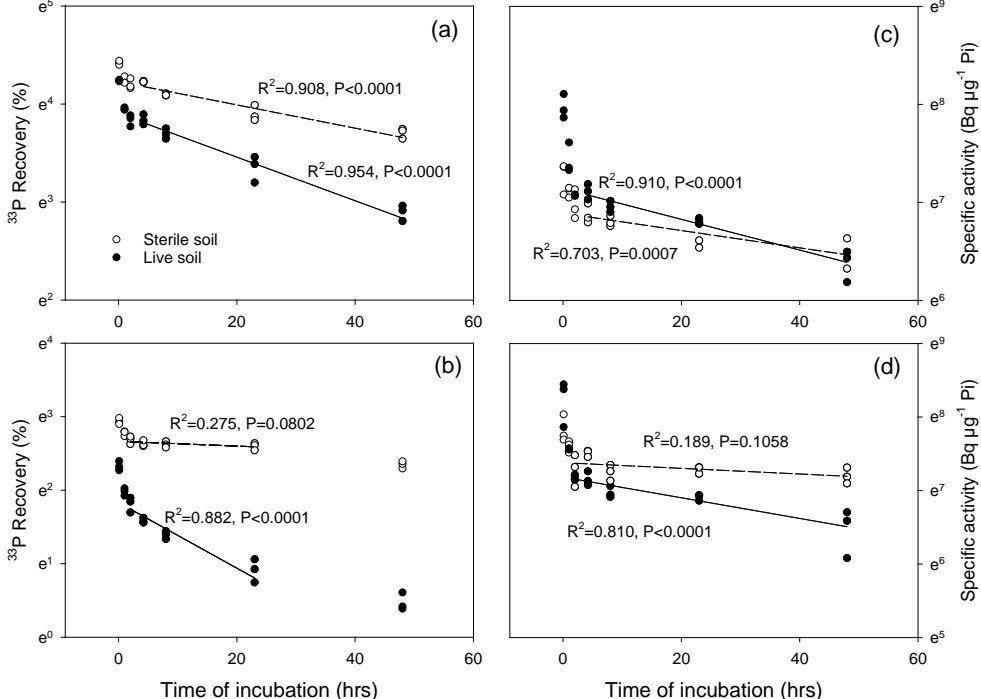




Figure 5. Net immobilization rates of $^{33}P_i$ by abiotic processes (sorption; A, C) and biotic processes (microbial uptake; B, D) measured for a temperate and a tropical soil after 0, 1, 2, 4, 8, 24 and 48 hours (A, B) and for six soils measured after 0, 4 and 24 hours (C, D). Temperate grassland soils (2, 4, 6) and tropical forest soils (3, 5, 7) were investigated. Curvilinear regressions following the function "exponential rise to maximum" were performed on the data in (A, B). Statistical analyses of data in (C, D) were run by two-way ANOVA for the factors soil and time (0, 4 and 24 hours after tracer addition), and the interaction of both factors.

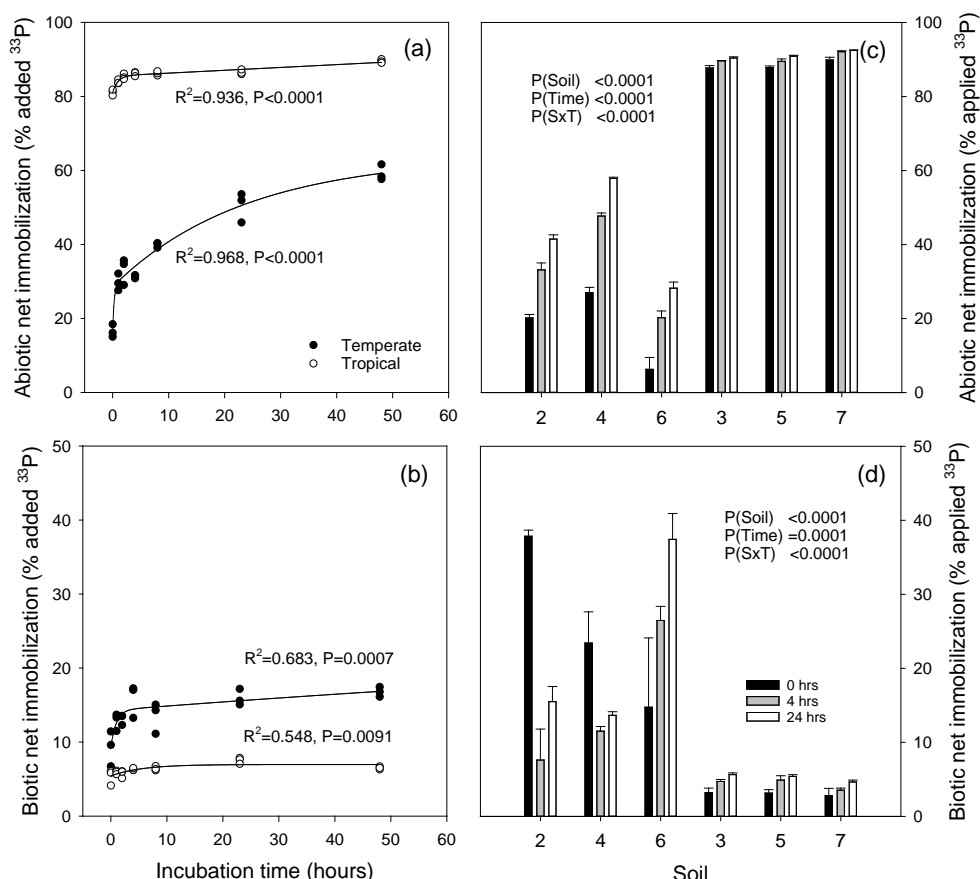






Figure 6. Gross influx rates into the available soil $P_i$ pool (A) and gross efflux rates from this pool (B) measured by $^{33}P$ isotope pool dilution for six soils over the time period 4 to 24 hours and assessed in sterile and live soils. Abiotic and biotic process rates are indicated by open and closed bars, respectively. Temperate grassland soils (2, 4, 6) and tropical forest soils (3, 5, 7) were studied. Presented are means ±1SD of triplicate live and sterile soils per time point and soil type. One-way ANOVA was performed on transformed data as indicated in brackets. Different lower case letters indicate significant differences between soils for abiotic processes (open bars), upper case letters for biological processes (black bars).

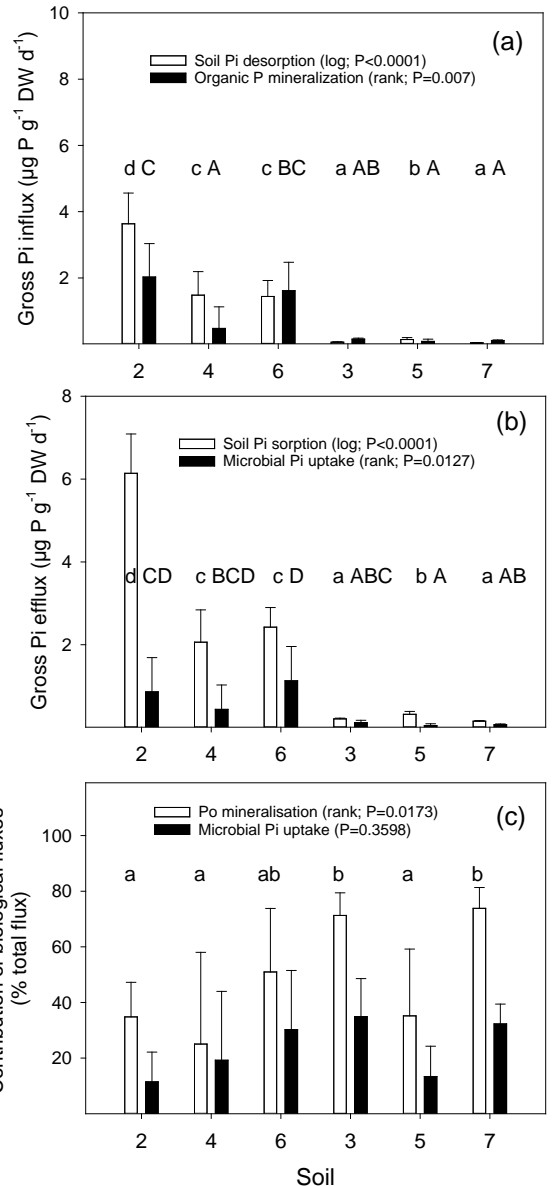



Figure 7. Relationship between selected soil physicochemical parameters, net abiotic and microbial immobilization fluxes, gross $P_i$ influx rates by biological processes (gross $P_o$ mineralization) and abiotic processes (gross $P_i$ desorption), and gross $P_i$ efflux rates by biological processes (gross microbial $P_i$ uptake) and abiotic processes (gross $P_i$ sorption). Regression lines are for linear or power function fits, and P and $R^2$ values for these are shown. Units are provided in Table 2 for soil physicochemical parameters and phosphomonoesterase, are % of added tracer for net processes, and μg P $g^{-1}$ soil dw $d^{-1}$ for gross process rates.

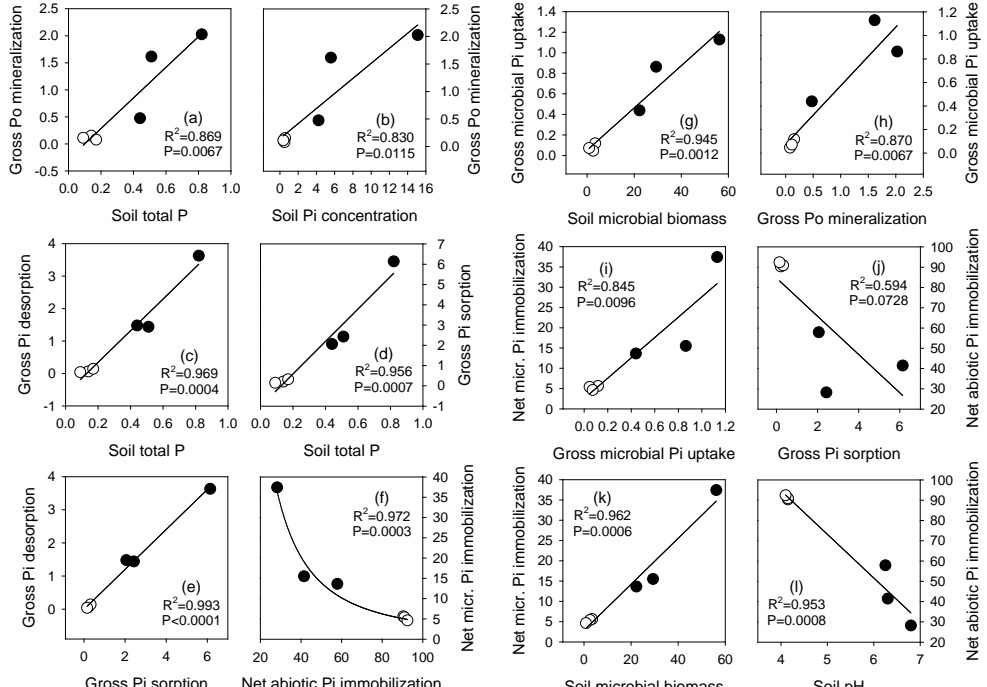