# Peer review of "A novel isotope pool dilution approach to quantify gross rates of key abiotic and biological processes in the soil phosphorus cycle"

_Biogeosciences, 2018_

## Referee Comment (RC1) · Anonymous Referee #1 · 5 Mar 2019

The manuscript presents an improved version of the isotope dilution approach to quantify gross P immobilization-mineralization rates and their control by abiotic and biotic processes. The originality of their approach is to apply the isotope dilution approach to sterile and non-sterile soils in order to separate the contribution of biotic and abiotic processes to P release or uptake. This is particularly important in the case of P since this element rapidly circulates in mineral and organic pools in soil. My feeling is that the experimental approach is solid: the improved proposed approach is compared to the classical/older one, the microbial immobilization of P quantified by the isotope dilution is checked with another method (extraction with chloroform), two methods of P extraction are compared and several soils are tested. Given the importance of P

for biological organisms and its coupling to N and C cycles, studies on P are highly desirable. The proposed method will facilitate the study of ecosystem P dynamics. Therefore I strongly support this manuscript. Below some comments with the aim of improving the manucript 1) The study includes combines two methods of P flux measurements, different extraction procedure, kinetics....it's not easy to follow. It might be useful to build a figure with an overview of all experimental works 2) It might be useful to specify as soon as the introduction that the method of Bartholomew was initially designed for N. Then this method was adopted for P. 3) A big limit of your approach: when you estimate the contribution of abiotic and biotic by calculating the difference of P immobilization (or mineralization) between sterile and non-sterile soils, you assume that biotic and abiotic P fluxes are additive. In reality, the two types of process interact, e.g. competition for soluble P. This question is rapidly evacuated which is not a good idea. More generally, you are too subjective when you consider the different limits of your approach. You always came to the conclusion that your method is perfect. It would be more relevant for the community to see where the main limits and possibilities of improvements for the future are? 4) I was surprised by the statement that most of the organic P is present in the microbial biomass. Is it a general result or specific to some soils? 5) The way you calculate the P fluxes with the dilution isotopic approach is clearly explained, especially because you present the equations used. This is not the case for the IEX/ID method. Could you make an effort to clarify this? 6) There is an interesting pattern that you do not discuss: in tropical soils the Pi is rapidly/mainly immobilized by minerals whereas the Pi is released from microoganisms. On the long run this functioning cannot be maintained since the microbial pool P will exhaust. In nature, plants might desorbs this Pi to re-inject it in the living/organic cycle. I suggest you to read the studies on the "bank mechanism" that has been initially developed for N but could work for P (Fontaine et al 2011; Shahzad et al 2012; Perveen et al, 2014...). A little conclusion on the ecological relevance of your findings might be welcome. 7) On the writing: try to shorten your discussions, some sections are too long and dilute your main message. For example, section 4.5 is very technic and we do not see immediately the relevance. Your sentences should not exceed three lines and certainly do not reach 5 lines.

---

## Referee Comment (RC2) · Anonymous Referee #2 · 6 Mar 2019

General comments With great interest, I have read the bg discussion paper by Wanek et al. entitled "A novel isotope pool dilution approach to quantify gross rates of key abiotic and biological processes in the soil phosphorus cycle". The paper aims to improve the isotope pool dilution approach to assess gross organic P mineralization rates that has been proposed by various authors (Kirkham and Bartholomew, 1954; Di et al. 2000; Randhawa et al. 2005, Kellogg et al. 2006). In principle, the study seems well executed. However, I would like to raise a couple of points that are rather critical and could be improved. My main comment is that the claimed novelty of the method lies in the introduction of an abiotic control. However, the adequacy of autoclaved soils is questionable. Therefore, further experiments would be needed to fully prove the

validity of the measured rates. A second worry is that the description of the experiment is not too clear. You probably need to add a scheme showing the procedure, and maybe detail also the recommended procedure at the end of the manuscript. Thirdly, I am disappointed that the suggested method was not compared to existing protocols, whether this concerns the use of water as an extractant, the sequential extraction of microbial P, or the comparison to the Po mineralization method by Oehl et al. For this, you could contact Klaus Jarosch who is at present using the method, or Astrid Oberson. For sure, they will be willing to collaborate and compare the two approaches on a few soils. If you include some additions of quickly mineralized compounds such as RNA, you should be able to evaluate both methods thoroughly. Finally, you do not mention a possibility to derive net organic P mineralization rates. Could you please address this in the discussion? In conclusion, the manuscript still needs some major improvements before it can be accepted.

Specific comments Line 10: this first sentence is a rather bold statement, or wouldn't net rates of soil processes suffice for modelling purposes? Line 56: IEX – this is a new abbreviation, at least I don't think I have seen it before. Most of the studies categorized as IEX studies used isotopic exchange kinetics (IEK) techniques. I wonder if the abbreviation is needed or if it would not be easier to speak of isotope exchange techniques throughout the text. The second abbreviation (ID) is rather confusing due to the similarity to IPD and is not needed at all, I think. Line 94-99: this whole argument is highly questionable. An incomplete extraction of a homogeneous pool is not a problem, because the specific activity in the pool is captured and enters the calculations. Line 104: need (not needs) Line 113: it appears to be quite risky to recommend only two time points, given that you state that the rates should be constant during the measurement period. Line 147: Determination of total soil P (and total soil Pi) by extraction of ignited and non-ignited soils with 0.5 M H2SO4 is not valid for highly weathered tropical soils (see Condron, L.M., Moir, J.O., Tiessen, H., Stewart, J.W.B., 1990. Critical evaluation of methods for determining total organic phosphorus in tropical soils. Soil Science Society of America Journal 54, 1261-1266.) Line 153 onwards: in my

experience, autoclaving is very effective to sterilize soils. But re-colonization has to be avoided by all means, since there is ample substrate available and no competition. To me, it is not entirely clear which measures were taken to keep abiotic controls sterile throughout the experiment. Line 211: "the procedure as outlined above": please avoid these unclear references – above I see a long description of the isobutanol procedure. Is this what you mean? Line 213: how can you do time point 0 h? Line 217: why do you use different volumes of 0.5 M NaHCO3 for different soils? Line 218: there is a problem with this sequential extraction, as you state yourself later in the manuscript: a second extraction in the absence of chloroform would release P... It is okay to test things during method development, but then you need to describe the recommended procedure at the end of the manuscript – if you have tested the method thoroughly and ideally compared it to existing protocols. Line 221: it would be better to detail the calculations that were made. Line 230: abiotic immobilization – at which time? I suppose you mean at a given time point, but it would help the reader that you write this (or give a proper equation). Line 234: ok, this is a rapid assessment, but is it precise? Is abiotic immobilization not affected by autoclaving? Line 241: "had to be corrected" – are you still referring to previous work, or do you want to announce that gross rates ... have to be corrected for...? Line 243: you need to state here that this calculation occurs under the assumption that abiotic influx is unaffected by autoclaving, i.e. abiotic influx in sterilized and live soils is similar. Line 289: an increase in available Pi by 1.6-1.9 will affect tracer behavior. E.g. Table 3 and Table 5 in Bünemann et al. 2007 give IEK values indicative of fast and slow Pi sorption as well as concentration of Pi in soil solution and E1min as an estimate of isotopically exchangeable = available P. Importantly, these autoclaved soils were incubated for several weeks before doing these assessment, thus, the first P flush after autoclaving has partly been resorbed by the soil. At the very least, you need to state clearly for how long autoclaved soils were kept until the experiments were performed, or if you once checked for sterility at the end of the experiments. Line 295-308: please state if this was done in live soils only, or if abiotic controls were included. line 312: Please state the number of replicates from which the detection limit

was calculated. Line 334-336: it would be useful to refer to Fig. 4 already after this sentence. Line 339-343: these observations are completely in line with isotope dilution theory (e.g. Fardeau, Frossard). Line 363: it would be very interesting to compare this net 33 P immobilization by microbes with calculations based on fumigation-extraction techniques of 33P-labeled soils. Line 371-375: how is this possible? I think you need to word more carefully here, since you are presenting results of an approach (sequential extraction) which you later state is probably not useful. Line 416: please indicate where Bünemann, 2015 suggested that the IPD approach necessitates constant pool sizes (see end of section 2.6 reading "the (erroneous) assumption of a constant pool Q in the Di approach is problematic") Line 425-428: I don't think the result was that clear with respect to E values. Line 469-472: differences between field replicates have completely different underlying reasons such as gradients or heterogeneity in the field, or fertilization effects, and thus this argumentation is not helping at all here. Line 473: how do you account for it if you calculate a difference between abiotic and live soils? Line 488: depends on what you mean by "closely"... Line 500: I am not sure what you mean with "is included with water-extractable P", at least in Bünemann et al. 2007 33P activity in microbial biomass was determined by difference between fumigated and non-fumigated samples. Line 610: yes, I would have expected to find this in the present manuscript. Consider maybe adding small amounts of organic P substrates that are known to mineralize quickly in soils and assess the rate changes using either method. Line 617: do you have evidence for organic P availability? I did not see it... Line 655: can gross Pi sorption rates exceed gross Pi desorption rates 2-fold without a significant change in pool sizes? Did you see the intermediate draw down of available Pi mentioned in line 658? Line 669-670: this well known and not a new finding (as you make it sound). Table 1 column IEX/ID: measure P pool: water-extractable Pi, sometimes including microbial P – yes, true, but microbial P is not determined for calculation of gross Po mineralization Table 2: why are the soils not named 1-6, or which soil is soil 1? Please state methods for total organic P and soil Pi in a footnote under the table. Figure 3: can you indicated temperate and tropical soils in a and b as well? Figure

4c: how can the specific activity be greater in live than in sterile soils? Figure 5: which temperate and which tropical soil were used here?; add "measured in sterile soils" after (sorption, A, C) and "measured in live soils" after (microbial uptake, B, D). Figure 7: explain the symbols (for tropical and temperate soils) in a legend or in the caption.

---

## Author Comment (AC1) · 30 Apr 2019

Response letter to reviews of Biogeosciences Discussion Paper bg-2018-519 Dear Editorial Board, dear reviewers, please find our detailed responses to the reviewers' comments below, numbered consecutively (#1-#52) and indicated in blue. Changes to the text in the revised manuscript are marked in bold, italics indicate references to the text of the first manuscript version. References to line numbers refer to the lines in the revised manuscript version (exception: response #8). Kind regards, Wolfgang Wanek Interactive comment on "A novel isotope pool dilution approach to quantify gross rates of key abiotic and biological processes in the soil phosphorus cycle" by

Wolfgang Wanek et al. Anonymous Referee #1 The manuscript presents an improved version of the isotope dilution approach to quantify gross P immobilization-mineralization rates and their control by abiotic and biotic processes. The originality of their approach is to apply the isotope dilution approach to sterile and non-sterile soils in order to separate the contribution of biotic and abiotic processes to P release or uptake. This is particularly important in the case of P since this element rapidly circulates in mineral and organic pools in soil. My feeling is that the experimental approach is solid: the improved proposed approach is compared to the classical/older one, the microbial immobilization of P quantified by the isotope dilution is checked with another method (extraction with chloroform), two methods of P extraction are compared and several soils are tested. Given the importance of P for biological organisms and its coupling to N and C cycles, studies on P are highly desirable. The proposed method will facilitate the study of ecosystem P dynamics. Therefore I strongly support this manuscript. Below some comments with the aim of improving the manuscript. #1: We greatly appreciate the kind support of our manuscript and incorporated all suggested changes. 1) The study includes/combines two methods of P flux measurements, different extraction procedure, kinetics....it's not easy to follow. It might be useful to build a figure with an overview of all experimental works. #2: The revised manuscript now provides a schematic overview of the final procedure as Figure 2. All other consecutive Figures therefore increased in their numbering by +1. 2) It might be useful to specify as soon as the introduction that the method of Bartholomew was initially designed for N. Then this method was adopted for P. #3: we changed the wording to "The isotope pool dilution approach (IPD) of Kirkham and Bartholomew (1954) was developed as a general tracer approach to measure gross rates of soil element cycle processes, but was most frequently applied to nitrogen cycling processes such as organic N mineralization and nitrification (Booth et al., 2005). This IPD approach can however also be transferred to measure gross rates of P cycle processes. It then also relies on the labelling of the Pi pool with $^{33}P$ or $^{32}P$ and on subsequent time-resolved measurements of concentrations and

specific activities of Pi (Table 1, Figure 1B)." see lines 79-84. 3) A big limit of your approach: when you estimate the contribution of abiotic and biotic by calculating the difference of P immobilization (or mineralization) between sterile and non-sterile soils, you assume that biotic and abiotic P fluxes are additive. In reality, the two types of process interact, e.g. compete for soluble P. This question is rapidly evacuated which is not a good idea. More generally, you are too subjective when you consider the different limits of your approach. You always came to the conclusion that your method is perfect. It would be more relevant for the community to see where the main limits and possibilities of improvements for the future are? #4a: Competition between biotic and abiotic processes: In this case we would overestimate abiotic process rates in autoclaved soils, due to lack of competition by biotic processes. This would effectively cause an underestimation of biotic processes i.e. organic P mineralization and microbial Pi uptake. We found some hint for this competition by increasing abiotic net immobilization (Fig. 6 in the revised MS) and increasing 33P contents of microbial biomass over the time period. We state now in lines 502-507: "However, the estimation of the contribution of abiotic and biotic processes is based on calculating the difference in P fluxes between sterile and non-sterile soils. This assumes that biotic and abiotic fluxes are additive while there is potential that both processes compete for available Pi. In this case we would overestimate abiotic process rates in autoclaved soils, due to lack of competition by biotic processes. This could effectively cause an underestimation of biotic processes i.e. organic P mineralization and microbial Pi uptake. To date we have no approach at hand to cope with this potential bias.". #4b: Limits and possibilities for improvement – we see the major limitations or possibilities for improvement in the abiotic controls, i.e. the autoclaving process. We now state that "Overall, there is therefore potential for method improvement, particularly in terms of using abiotic controls circumventing autoclaving (e.g. bacteriozides combined with phosphomonoesterase inhibitors) or correcting for autoclave-induced changes in aggregation and in soil Pi content." in lines 507-509. #4b.1. Using a combination of phosphatase inhibitors with sodium azide or HgCl2 that curtail microbial activity

might be an option to circumvent autoclaving, which is currently the only one option to successfully kill microbes and extracellular enzymes. In the previous version we wrote "While application of phosphatase inhibitors might be another viable option, we are only aware of one study testing this; application of silver nanoparticles to soils showed a general inhibitory effect on soil enzymes (Shin et al., 2012). Previous tests in our laboratory with two commercial phosphatase inhibitor cocktails (Sigma-Aldrich) at 10-fold the recommended final concentration did not significantly decrease IPD rates in two soils (data not shown), indicating an insufficient inhibition of extracellular phosphatases." in lines 462-467. To this we added "However, more rigorous tests of soil enzyme activities with synthetic substrates (e.g. MUF-Pi) and of P mineralization based on 33P-IPD using increasing concentrations and different types of commercial phosphatase inhibitor cocktails might make clear whether this approach is viable or not." In lines 467-469. #4b.2. We wrote already in the first version of this manuscript that "It must be noted that autoclaving could potentially alter the physicochemical properties of soils, thereby affecting abiotic sorption-desorption kinetics. Despite this, in previous studies autoclaving up to two times and steam sterilization did neither affect the cation exchange capacity, nor base saturation, soil surface area, contents of total organic carbon and total nitrogen, and only slightly soil pH (Wolf et al., 1989;Tanaka et al., 2003;Serrasolsas and Khanna, 1995b)." in lines 478-482. Sterilization by autoclaving might however somewhat deteriorate soil aggregation and therefore increase the number of sites available for sorption and desorption that were previously hidden in aggregates. We could however not find any reference showing this clearly. We added "Autoclaving might however weaken soil aggregates and therefore increase the number of sites accessible for sorption-desorption processes that were previously hidden in aggregates. However, we did not find clear support for or against this in the literature as autoclaving only weakly affected soil aggregate size distribution, causing a 0.5 to 1% increase in clay-sized compared to silt-sized aggregates (Berns et al., 2008). In contrast, aggregate stability and aggregation increased upon autoclaving in two other studies (Lotrario et al., 1995;Salonius et al., 1967). Effects of autoclaving

on soil aggregation and soil P dynamics could be tested by measuring P processes rates on intact aggregates <2 mm and after destroying them by ultrasonication or grinding." in lines 482-489. #4b.3. As autoclaving causes a flush of Pi from lysis of microbial biomass (see lines 489-494), this might cause a stimulation of soil Pi sorption and decrease soil Pi desorption. The latter would cause an overestimation of organic P mineralization. This was of minor importance as in some later experiments we experienced systematically higher, not lower Pi desorption rates (abiotic influx) that caused negative Po mineralization rates. We added "Nonetheless, effects of increased Pi mobilization due to microbial lysis on Pi sorption-desorption could be tested in sterile soils by adding increasing concentrations of non-labelled Pi alongside the 33Pi tracer and thereby could be corrected for in future 33P-IPD experiments." in lines 498-500. 4) I was surprised by the statement that most of the organic P is present in the microbial biomass. Is it a general result or specific to some soils? #5: This has really been found e.g. in the Franz Josef Glacier primary succession sequence (Turner et al. 2013). To shorten the manuscript we deleted this part. 5) The way you calculate the P fluxes with the isotopic dilution approach is clearly explained, especially because you present the equations used. This is not the case for the IEX/ID method. Could you make an effort to clarify this? #6: We added an extensive description of the equations and explain the terms used in the IEK approach. See lines 57-78: "In short-term IEK experiments abiotic sorption/desorption processes from an isotopically exchangeable Pi pool are measured over a short time period in batch experiments (100 min, 1:10 (w:v) soil: water slurry, $\pm$ microbicides), assuming that no microbial tracer uptake occurs (Table 1). In such short-term IEK experiments the decrease in radioactivity (radiotracer recovery) in soil water is described by a power function: $r(t)/R = r_{1min}/R \times t^{-n}$ R is the added radioactivity and $r(t)$ the radioactivity recovered at any time $t$ in soil water extracts. The parameters $r_{1min}/R$ and $n$ (slope of the regression indicating speed of isotopic exchange) are derived from the log-log regression of $r(t)$ versus time. This is based on steady state assumptions, i.e. that Pi concentration in soil water extracts ($C_P$) is constant. In some soils an extended version of this equation needs

to be applied: r(t)/R = m x (t + m1/n)-n + rinf/R Here, rinf/R is the maximum possible dilution of the added radiotracer, approximated as the ratio of CP to total inorganic P in soils. n and m are derived from non-linear fitting procedures. Assuming that tracer and tracee behave similarly in the system, the specific activity of Pi in soil solution should reflect the specific activity of isotopically exchangeable P – termed E-value (in mg P kg-1 soil). E(t) = CP / (r(t)/R). Isotopic dilution (E'(t)) is further measured over the full length of a moist soil incubation experiment lasting for several days to weeks, constituting the total amount of exchangeable Pi or isotope dilution caused by concurrent biological processes (Po mineralization) and physicochemical processes. Short-term exchange kinetics by non-biological processes are then extrapolated over the full time period of the moist soil incubation (E(t)). The difference between E'(t) and the extrapolated E(t)-value provides then the measure of gross Po mineralization."
6) There is an interesting pattern that you do not discuss: in tropical soils the Pi is rapidly/mainly immobilized by minerals whereas the Pi is released from microorganisms. On the long run this functioning cannot be maintained since the microbial pool P will exhaust. In nature, plants might desorb this Pi to re-inject it in the living/organic cycle. I suggest you to read the studies on the "bank mechanism" that has been initially developed for N but could work for P (Fontaine et al 2011; Shahzad et al 2012; Perveen et al, 2014...). A little conclusion on the ecological relevance of your findings might be welcome. #7: Thank you for this suggestion. We added the following in lines 673-682: "The observed rates indicate that soils immobilized more Pi than they mobilized by abiotic processes, causing an intermediate draw-down of available Pi pools Two processes work against this draw down of Pi in soils, i.e. Po mineralization and microbial Pi release through turnover and lysis. Moreover, plants (and microbes) might also desorb this sorbed Pi by release of phytosiderophores and organic acids and thereby replenish Pi and re-inject it in the organic P cycle. Similar to the soil C-N cycle we might also expect an active "bank mechanism" regulating nutrient and C sequestration in soils . At high nutrient availability priming effects are low, allowing the sequestration of nutrients and SOC build-up. At low nutrient availability microbes

(and plants) release nutrients from SOM and mineral surfaces stimulated by root exudates, effectively mining inorganic and organic P stored in soils." 7) On the writing: try to shorten your discussions, some sections are too long and dilute your main message. For example, section 4.5 is very technic and we do not see immediately the relevance. Your sentences should not exceed three lines and certainly do not reach 5 lines. #8: we deleted several parts of the discussion, i.e. referring to the first version of the manuscript we deleted lines 401-402, 427-428, 434-437, 448-451, 462-466, 469-473, 500-506, 586-595, 607-610, 664-668, 672-676, and shortened overly long sentences. Anonymous Referee #2 General comments With great interest, I have read the BG Discussion paper by Wanek et al. entitled "A novel isotope pool dilution approach to quantify gross rates of key abiotic and biological processes in the soil phosphorus cycle". The paper aims to improve the isotope pool dilution approach to assess gross organic P mineralization rates that has been proposed by various authors (Kirkham and Bartholomew, 1954; Di et al. 2000; Randhawa et al. 2005, Kellogg et al. 2006). In principle, the study seems well executed. However, I would like to raise a couple of points that are rather critical and could be improved. #9: We are highly grateful for the overall positive response and have revised the manuscript according to the critical comments, as provided below. My main comment is that the claimed novelty of the method lies in the introduction of an abiotic control. However, the adequacy of autoclaved soils is questionable. Therefore, further experiments would be needed to fully prove the validity of the measured rates. #10: we are fully aware that autoclaving soils bears potential disadvantages such as potentially disturbing soil aggregation (little support found for this) and releasing microbial P as a flush while other major physicochemical properties of soils are not affected e.g. SOC and texture. For more critical thoughts please see our response to comment #4b.2 by reviewer 1. But most importantly for this study autoclaving curtails extracellular phosphatase activity (!) and kills soil microbes and this has not been done so far by any other IPD or IEK approach, as adding $HgCl_2$ only inhibits microbial but not phosphatase activity. Therefore the autoclaved/sterile soils are key to allow

measurements of abiotic rates. A second worry is that the description of the experiment is not too clear. You probably need to add a scheme showing the procedure, and maybe detail also the recommended procedure at the end of the manuscript. #11: As outlined in our response to comment #2 the revised manuscript now provides a schematic overview of the final procedure as Figure 2. Thirdly, I am disappointed that the suggested method was not compared to existing protocols, whether this concerns the use of water as an extractant, the sequential extraction of microbial P, or the comparison to the Po mineralization method by Oehl et al. For this, you could contact Klaus Jarosch who is at present using the method, or Astrid Oberson. For sure, they will be willing to collaborate and compare the two approaches on a few soils. If you include some additions of quickly mineralized compounds such as RNA, you should be able to evaluate both methods thoroughly. #12: We are fully aware of this short-coming and mentioned this in lines 627-629 "A direct comparison of the present IPD and the IEK approaches on the same soils might help to clarify how far the approaches really deviate in their gross Po mineralization rate estimations.". However, this study just reports the method development and demonstrates the validity of the newly adopted IPD approach. In a future cooperation with the group of Emile Frossard and Astrid Oberson we shall directly compare both approaches on the same set of soils. First communications have been started between Wolfgang Wanek and Astrid Oberson in that respect. Addition of labile organic P – as suggested by the reviewer – would object the steady state assumption underlying the IEK approach. This would therefore not allow the application of the IEK approach, while the IPD approach can cope with changing Pi concentrations. Finally, you do not mention a possibility to derive net organic P mineralization rates. Could you please address this in the discussion? #13: Net organic P mineralization rates can easily be derived by subtracting gross microbial Pi uptake from gross Po mineralization rates. We indicate this now in lines 276-277, stating "Net organic P mineralization rates can easily be derived by subtracting gross microbial Pi uptake from gross Po mineralization rates". In conclusion, the manuscript still needs some major improvements before it can be accepted. Specific comments

Line 10: this first sentence is a rather bold statement, or wouldn't net rates of soil processes suffice for modelling purposes? #14: we deleted gross rates and now just state that "…are limited by the availability of global data on rates of soil P processes" – line 10-11. Line 56: IEX – this is a new abbreviation, at least I don't think I have seen it before. Most of the studies categorized as IEX studies used isotopic exchange kinetics (IEK) techniques. I wonder if the abbreviation is needed or if it would not be easier to speak of isotope exchange techniques throughout the text. The second abbreviation (ID) is rather confusing due to the similarity to IPD and is not needed at all, I think. #15: we exchanged all IEX/ID abbreviations to IEK in the manuscript and in the Tables. Line 94-99: this whole argument is highly questionable. An incomplete extraction of a homogeneous pool is not a problem, because the specific activity in the pool is captured and enters the calculations. #16: An incomplete extraction of a homogenous exchangeable Pi pool affects IPD rates two-fold, (i) causing influx by desorption of unlabelled Pi which violates assumption 2, and (ii) underestimation of this exchangeable Pi pool - even if specific activities thereof are correctly measured - also causes underestimation of IPD rates given that Pi concentration linearly affects IPD rates according to equation 1 and 2 in lines 272-273. In lines 105-111 we stated that "The soil extraction should target the bio-available exchangeable Pi pool. Pi in soil solution, however, undergoes rapid equilibration with easily adsorbed Pi. An incomplete extraction of this pool causes an underestimation of Po mineralization rates, due to desorption from this pool, causing an influx of unlabeled tracer (together with unlabeled Pi) into the target pool, and thus violates assumption #2 of IPD assays. The commonly used soil water extractions target only a small fraction of this target pool, whereas standard soil P extractants, such as Olsen, Mehlich-3 or Bray-1, extract a larger fraction (Kleinman et al., 2001) and, therefore, are suggested to be better suited to extract the rapidly exchanging Pi pool (Kellogg et al., 2006)." Moreover in lines 512-532 we stated that "Similar to 15N IPD assays, where salt extractions are employed to target the available inorganic or organic N pool (Murphy et al., 2003;Wanek et al., 2010;Hu et al., 2017), we focused on the potentially bio-available,

salt-extractable Pi pool that reflects the plant- (and microbial) accessible amount of soil Pi better (Fardeau et al., 1988;Olsen et al., 1954;Horta and Torrent, 2007) than the water extractable Pi pool that is mostly assessed with soil IEK methods. ..... Several studies compared soil P tests like Bray III, resin P, and Olsen-P to soil water Pi and plant P uptake in order to assess how well they reflect the available Pi pool. These studies demonstrated that soil tests like bicarbonate extractions (Olsen-P), resin P and DGT (diffusive gradients in thin films technique) closely resembled the SA values of Pi extracted by water or 10 mM $CaSO_4$ or from plants (Six et al., 2012;Fardeau et al., 1988;Demaria et al., 2005). ... bicarbonate extracted 8- to 22-fold greater amounts of exchangeable Pi compared to water and SA of Pi in bicarbonate extracts reached 66-90% of the SA values measured in soil water extracts (Demaria et al., 2005). IPD approaches require fast extractions to quickly terminate the assay after 4 and 24 h, which renders water extractions (generally 16 h), resin P (16 h) and DGT (up to 48 h in low P soils; (Six et al., 2012)) impossible. Bicarbonate extractions only take 30-60 min and therefore represent a viable alternative. Moreover, it makes the IPD assay on average 8-fold more sensitive as a greater amount of exchangeable Pi is extracted by bicarbonate than with water (Kleinman et al., 2001).". We added "Underestimation of this labile Pi pool - even if specific activities thereof are correctly measured - also causes underestimation of IPD rates given that Pi concentrations linearly affect IPD rates according to IPD equations 1 and 2" in lines 533-535. Line 104: need (not needs) #17: done Line 113: it appears to be quite risky to recommend only two time points, given that you state that the rates should be constant during the measurement period. #18: We changed the sentence accordingly (lines 125-128) "The minimum two time points necessary to measure concentration and specific activity of Pi for the IPD calculations should therefore lie in between the initial phase and the start of re-mineralization but it is recommendable to test more time points in the beginning to test time linearity of IPD rates for specific soil types." Line 147: Determination of total soil P (and total soil Pi) by extraction of ignited and non-ignited soils with 0.5 M $H_2SO_4$ is not valid for highly weathered tropical soils (see Condron,

L.M., Moir, J.O., Tiessen, H., Stewart, J.W.B., 1990. Critical evaluation of methods for determining total organic phosphorus in tropical soils. Soil Science Society of America Journal 54, 1261-1266.) #19: we added the following in lines 164-166: "We must however surmit that ignition methods tend to overestimate soil organic P in highly weathered tropical soils (Condron et al., 1990)." Line 153 onwards: in my experience, autoclaving is very effective to sterilize soils. But re-colonization has to be avoided by all means, since there is ample substrate available and no competition. To me, it is not entirely clear which measures were taken to keep abiotic controls sterile throughout the experiment. #20: The autoclaving was performed 48 and 2 hours before starting the IPD experiments. The very close time point of autoclaving and start of IPD experiments therefore completely hindered microbial recolonization. Moreover, we saw that enzyme activities (FDA hydrolysis as proxy for microbial active biomass) and phosphomonoesterase further declined towards 24 and 48 hours after the 2nd autoclaving cycle, indicating that microbes did not recolonize during that time frame. Lines 308-315 we stated this "Our results show that two consecutive treatments of the soils by autoclaving, with a 48 hours incubation in between, effectively reduced microbial metabolic activity as shown by the reduction in soil FDA hydrolysis by 90% in soil 4 and by 97-99% in all other soils (Fig. 2). Autoclaved soils did not show any increase in soil microbial activity during the two days of incubation. On the contrary, the inhibition of FDA hydrolysis even increased from 1 hour (all soil average: 94%) towards 24 and 48 hours after sterilization (average: 97-99%). The inhibition of extracellular acid phosphatase activity was almost complete in tropical soils (95-97%) and strongly reduced in temperate soils (79-80%). Similar to FDA hydrolysis the extent of inhibition of phosphatase activity increased from day 0 (average: 86%) to day 1 and 2 (average: 88-89%, Fig. 2)." Line 211: "the procedure as outlined above": please avoid these unclear references – above I see a long description of the isobutanol procedure. Is this what you mean? #21: deleted Line 213: how can you do time point 0 h? #22: by adding the tracer solution and immediately extracting the soils afterwards. Line 217: why do you use different volumes of 0.5 M NaHCO3 for different soils? #23: We added

the following explanatory text in lines 185-187 "Lower extractant volumes in tropical and other P poor soils are used to reach higher Pi concentrations in the bicarbonate extracts for better quantification." Line 218: there is a problem with this sequential extraction, as you state yourself later in the manuscript: a second extraction in the absence of chloroform would release P. . . It is okay to test things during method development, but then you need to describe the recommended procedure at the end of the manuscript – if you have tested the method thoroughly and ideally compared it to existing protocols. #24: In this study we only tested the sequential i.e. sECE procedure which works well for microbial C and N. Unfortunately we only later realized that the nature of the exchangeable and extractable Pi pool is soils is very different compared to soil inorganic and organic N and therefore did not perform a method comparison e.g. with chloroform-fumigation extraction as traditionally used. We clearly state the inadequacy of sECE and recommend CFE as mentioned in lines 384-389: "Sequential extraction-liquid chloroform-extraction (sECE) was applied to directly follow net 33P uptake by microbes, whereas biotic net 33P immobilization was estimated indirectly as the difference in net 33P immobilization by live and sterile soils. In the two measured soils, sECE estimates of microbial net 33P uptake were higher than the microbial net 33P immobilization estimates (temperate soil: 24.6% vs. 16.0%, and tropical soil: 16.8% vs. 7.5%, for direct and indirect estimates, respectively). This indicates incomplete extraction of exchangeable Pi prior to microbial lysis with chloroform." and in lines 558-570 "Repeated extractions with bicarbonate also showed that the first extraction only removed 67-78% of the 33Pi that was extractable with three consecutive extractions (D. Wasner, data not shown). . .. Delineation of the causes could be performed by a comparison of sECE with liquid chloroform-fumigation extraction (CFE) i.e. parallel assessments of microbial 33P uptake, using a comparison of 33P in bicarbonate versus bicarbonate+liquid chloroform or bicarbonate+liquid hexane extracts. Given the continued extraction of Pi from exchangeable Pi pools in serial extraction tests, parallel determination of microbial P and 33P by CFE seems favorable relative to sequential extractions for microbial P determination.". Line 221: it would be better to detail the

calculations that were made. #25: We added the following in lines 241-243: "Volume corrections were calculated as soil wet weight after centrifugation minus fresh weight weighed into each tube in grams, divided by the density of the bicarbonate solution (in g/mL), providing the carry-over of extractant from the first extraction in mL." Line 230: abiotic immobilization – at which time? I suppose you mean at a given time point, but it would help the reader that you write this (or give a proper equation). #26: the information was added i.e. after 0, 1, 2, 4, 8, 24 and 48 hours (see 253, and Figure 6), i.e. 0, 1, 2, 4, 8, 24 and 48 hours in the time kinetic experiment and after 0, 4 and 24 hrs in the six-soil comparison experiment. Line 234: ok, this is a rapid assessment, but is it precise? Is abiotic immobilization not affected by autoclaving? #27: The effect of autoclaving on abiotic processes is discussed at length in #4b.2 and #4b.3. Relevant discussion was added to the discussion as also outlined in these two responses. Line 241: "had to be corrected" – are you still referring to previous work, or do you want to announce that gross rates . . . have to be corrected for. . . ? #28: This refers to the present study. We changed to make it more concise in lines 261-264: "In these previous studies abiotic processes were not corrected from the final data. However, to calculate gross Po mineralization, gross rates of Pi desorption have to be corrected for in live soils. This is due to gross rates of Pi influx in live soils being the integral of abiotic processes (desorption) and biological processes (Po mineralization)." Line 243: you need to state here that this calculation occurs under the assumption that abiotic influx is unaffected by autoclaving, i.e. abiotic influx in sterilized and live soils is similar. #29: In the revised manuscript we added in lines 269-271: "Both abiotic corrections are based on the assumption that abiotic sorption/desorption processes are not affected by autoclaving, i.e. that these processes act similarly in sterile and in live soils." Line 289: an increase in available Pi by 1.6-1.9 will affect tracer behavior. E.g. Table 3 and Table 5 in Bünemann et al. 2007 give IEK values indicative of fast and slow Pi sorption as well as concentration of Pi in soil solution and E1min as an estimate of isotopically exchangeable = available P. Importantly, these autoclaved soils were incubated for several weeks before doing these assessment, thus, the first P

flush after autoclaving has partly been resorbed by the soil. At the very least, you need to state clearly for how long autoclaved soils were kept until the experiments were performed, or if you once checked for sterility at the end of the experiments. #30: We clearly stated that "Soils were then either sterilized twice, 48 and 2 h before start of the IPD experiments, by autoclaving at 121 °C for 60 min (sterile soils), or were kept at 20 °C (live soils)." in lines 171-172. Soils were therefore used only 2 hours after the second autoclaving cycle which hinders microbial recolonization. See also comment #20. The IEK data in Bünemann et al. 2007 for five soils after three times autoclaving and a following pre-incubation of sterile soils for four weeks showed that r/R (tracer recoveries) values were greater in sterile than in live soils due to killing microbes and therefore inhibiting microbial tracer immobilization. The isotopic dilution rate "n" was higher in live than in sterile soils due to faster immobilization by abiotic and biotic mechanisms, and CP (Pi contents) were greater in sterile than in live soils, due to the killing of microbes causing a flush of labile P. In contrast, E-values did not show a clear pattern. Overall, the patterns between live and sterile soils were very similar to our data see e.g. differences in slopes between live and sterile soils in Figure 5. Line 295-308: please state if this was done in live soils only, or if abiotic controls were included. #31: We added this information in line 321-322 ", including both live and sterile soils". Line 312: Please state the number of replicates from which the detection limit was calculated. #32: We added this information in line 340 "of gross Po mineralization measured for the three tropical soils (each measured in triplicates)". Line 334-336: it would be useful to refer to Fig. 4 already after this sentence. #33: done Line 339-343: these observations are completely in line with isotope dilution theory (e.g. Fardeau, Frossard). #34: true Line 363: it would be very interesting to compare this net 33P immobilization by microbes with calculations based on fumigation-extraction techniques of 33P-labeled soils. #35: Unfortunately in these experiments we did not perform CFE alongside sECE and net microbial immobilization measurements. See also response #36. But this is planned during one of the next measurement campaigns. Line 371-375: how is this possible? I think you

need to word more carefully here, since you are presenting results of an approach (sequential extraction) which you later state is probably not useful. #36: See rewording as described in response #24. We now recommend to use CFE instead of EFE. Line 416: please indicate where Bünemann, 2015 suggested that the IPD approach necessitates constant pool sizes (see end of section 2.6 reading "the (erroneous) assumption of a constant pool Q in the Di approach is problematic") #37: We are sorry for this mistake i.e. it was the Di-IPD approach as applied e.g. by Randhawa et al. that wrongly applied steady state assumptions. The IPD approach developed here is not based on steady state assumptions and can easily accommodate changing Pi concentrations. We changed the wording in lines 440-443 as follows: "The equations to estimate IPD rates can easily be solved for soils where target pool concentrations increase (net mineralization) or decrease (net immobilization) over time and where microbial biomass P changes (Kirkham and Bartholomew, 1954), and do not necessitate constant pool sizes as wrongly suggested previously (Di et al., 2000;Randhawa et al., 2005).". Line 425-428: I don't think the result was that clear with respect to E values. #38: True for r/R values (r/R values of 1/5 soils were altered) but Table 4 shows that E values differed in 3/5 soils (suspension > moist incubations) and CP in 4/5 cases (suspension > moist incubations).Wee changed "clearly demonstrated" to "indicated" in line 452. Line 469-472: differences between field replicates have completely different underlying reasons such as gradients or heterogeneity in the field, or fertilization effects, and thus this argumentation is not helping at all here. #39: we deleted this sentence. Line 473: how do you account for it if you calculate a difference between abiotic and live soils? #40: Please see our response #4b.3. We added additional text in line 498-500 "Nonetheless, effects of increased Pi mobilization due to microbial lysis on Pi sorption-desorption could be tested in sterile soils by adding increasing concentrations of non-labelled Pi alongside the 33Pi tracer and then could be corrected for in future 33P-IPD experiments." and in lines 502-509 "However, the estimation of the contribution of abiotic and biotic processes is based on calculating the difference in P fluxes between sterile and non-sterile soils. This assumes that biotic

and abiotic fluxes are additive while there is potential that both processes compete for available Pi. In this case we would overestimate abiotic process rates in autoclaved soils, due to lack of competition by biotic processes. This could effectively cause an underestimation of biotic processes i.e. organic P mineralization and microbial Pi uptake. To date we have no approach at hand to cope with this potential bias. Overall, there is therefore potential for method improvement, particularly in terms of using abiotic controls circumventing autoclaving (e.g. bacteriozides combined with phosphomonoesterase inhibitors) or correcting for autoclave-induced changes in aggregation and in soil Pi content." Line 488: depends on what you mean by "closely". . . #41: We were citing the paper of Demaria et al. 2005 here, stating that 520-528 "Several studies compared soil P tests like Bray III, resin P, and Olsen-P to soil water Pi and plant P uptake in order to assess how well they reflect the available Pi pool. These studies demonstrated that soil tests like bicarbonate extractions (Olsen-P), resin P and DGT (diffusive gradients in thin films technique) closely resembled the SA values of Pi extracted by water or 10 mM CaSO4 or from plants (Six et al., 2012;Fardeau et al., 1988;Demaria et al., 2005). . . . Bicarbonate extracted 8- to 22-fold greater amounts of exchangeable Pi compared to water and SA of Pi in bicarbonate extracts reached 66-90% of the SA values measured in soil water extracts (Demaria et al., 2005).". Line 500: I am not sure what you mean with "is included with water-extractable P", at least in Bünemann et al. 2007 33P activity in microbial biomass was determined by difference between fumigated and non-fumigated samples. #42: We deleted this sentence (lines 500-506, first MS version) and reference to microbial biomass in Table 1. Line 610: yes, I would have expected to find this in the present manuscript. Consider maybe adding small amounts of organic P substrates that are known to mineralize quickly in soils and assess the rate changes using either method. #43: see response #12 on the same topic. Line 617: do you have evidence for organic P availability? I did not see it. . . #44: no, unfortunately we did not measure other proxies of Po availability than total soil Po. We deleted and rewrote the sentence in lines 633-635 "This indicates that gross Po mineralization might rather be driven by

total P than by soil enzyme activity, and that total soil Po does not well represent the Po fraction accessible to soil phosphatases.". Line 655: can gross Pi sorption rates exceed gross Pi desorption rates 2-fold without a significant change in pool sizes? Did you see the intermediate draw down of available Pi mentioned in line 658? #45: Yes, this can happen in the short-term (days) but not over prolonged time periods as this would deplete the labile Pi pool completely. We saw a slight decrease in extractable Pi across the 24 hrs incubation period. We added the following text to better explain our findings in lines 672-682 "Gross Pi sorption rates exceeded gross Pi desorption rates approximately 2-fold but both were strongly related, indicating close and rapid cycling of available Pi through sorption-desorption processes. The observed rates indicate that soils immobilized more Pi then they mobilized by abiotic processes, causing an intermediate draw down of available Pi pools. Two processes work against this draw down of Pi in soils, i.e. Po mineralization and microbial P release through turnover and lysis. Moreover, plants (and microbes) might also desorb this sorbed Pi by release of phytosiderophores and organic acids and thereby replenish Pi and re-inject it in the organic P cycle. Similar to the soil C-N cycle we might also expect an active "bank mechanism" regulating nutrient and C sequestration in soils (Fontaine et al., 2011). At high nutrient availability priming effects are low, allowing the sequestration of nutrients and SOC build-up. At low nutrient availability microbes (and plants) release nutrients from SOM and from mineral surfaces stimulated by root exudates, effectively mining inorganic and organic P stored in soils." Line 669-670: this well-known and not a new finding (as you make it sound). #46: We rewrote this sentence to state "The strong negative relation between abiotic net Pi immobilization and soil pH re-confirms that strongly weathered, acid tropical soils have a higher P sorption and fixation capacity than temperate soils." lines 689-691. Table 1 column IEX/ID: measure P pool: water-extractable Pi, sometimes including microbial P – yes, true, but microbial P is not determined for calculation of gross Po mineralization. #47: We deleted microbial P from the lines of measured P pools for both approaches, IEK and IPD. Table 2: why are the soils not named 1-6, or which soil is soil 1? Please state methods for total organic

P and soil Pi in a footnote under the table. #48: This derives from historical reasons when starting soil sampling and selection. From this onwards the soils became their numbers and these were carried through the whole study. To better or easier link data and results to measurements we would like to keep this odd numbering. Otherwise we would need to renumber all Figures, Tables and particularly the data tables. Figure 3: can you indicate temperate and tropical soils in a and b as well? #49: Yes, the revised Figure 4 a and b now also shows temperate soils in closed symbols and tropical soils in open symbols. Figure 4c: how can the specific activity be greater in live than in sterile soils? #50: We added the same amount of radiotracer to live and sterile soils, but sterile soils had higher extractable Pi concentrations (caused by microbial P flush), causing lower SA of sterile and higher SA in live soils, during the earliest incubation phases. Figure 5: which temperate and which tropical soil were used here?; add "measured in sterile soils" after (sorption, A, C) and "measured in live soils" after (microbial uptake, B, D). #51: We changed the text accordingly i.e. "Net immobilization rates of 33Pi by abiotic processes (sorption; A, C) measured in sterile soils and biotic processes (microbial uptake; B, D) measured in live soils of a temperate grassland (soil 4) and a tropical forest (soil 3) after 0, 1, 2, 4, 8, 24 and 48 hours (A, B)". Figure 7: explain the symbols (for tropical and temperate soils) in a legend or in the caption. #52: We added the following to the Figure legend "Open circles depict tropical forest soils and closed circles temperate grassland soils.".

Please also note the supplement to this comment:
https://www.biogeosciences-discuss.net/bg-2018-519/bg-2018-519-AC1-supplement.pdf

---

## Author Comment (AC5) · 5 Jun 2019

Please find the revised version of the manuscript including a detailed point-by-point response to all comments by the two reviewers in the upload. Kind regards, Wolfgang Wanek

---

## Author Response (AR2)

Dear Editorial Board, dear Reviewer 2,

please find our point by point response to the reviewer's suggestions below. Our text is in blue, sentences added or changed in the manuscript in bold. We hope with these changes the manuscript now is acceptable for publication in Biogeosciences. We also provide a tracked manuscript version alongside the revised manuscript.

Kind regards,

Wolfgang Wanek

bg-2018-519r1

Anonymous Referee #2 Report #2 - Submitted on 21 Jun 2019

For final publication, the manuscript should be

    accepted subject to minor revisions

Suggestions for revision or reasons for rejection (will be published if the paper is accepted for final publication)

Thanks for addressing all comments and improving the manuscript.

I went through the answers and would like to comment on a few of them (using the numbers of the questions by reviewer 1 and subsequently your #:

1) this new figure is certainly helpful. Only the reason for the parallel steps acidification vs. isobutanol fractionation does not become clear. Maybe you could indicate what is derived from the comparison of the two ways of analysis?

Direct acidification followed by malachite green measures soil extract Pi concentration but LSC measures $^{33}P_i$ and $^{33}P_o$. In contrast isobutanol fractionation isolates Pi and therefore allows the measurement of concentrations and 33P activity in pure Pi (no $^{33}P_o$ interference). However, since there was no $^{33}P_o$ formation the results of both approaches were similar. We added the following to the legend of Figure 2: "**Isobutanol fractionation separates dissolved $P_i$ from $P_o$ and thereby allows highly specific measurements of concentrations and $^{33}P$ activities in $P_i$, without interference by $^{33}P_o$. Direct acidification of bicarbonate extracts measures dissolved $P_i$ using malachite green but LSC quantifies the sum of $^{33}P_i$ and $^{33}P_o$, the formation of the latter ($^{33}P_o$) however turned out to be insignificant.** " see lines 1119-1123.

2) Fine, although here you could maybe refer to Di et al. 2000

We added the reference to the following sentence "**The isotope pool dilution approach (IPD) of Kirkham and Bartholomew (1954) was developed as a general tracer approach to measure gross rates of soil element cycle processes, but was most frequently applied to nitrogen cycling processes such as organic N mineralization and nitrification (Booth et al., 2005). The IPD approach can however also be transferred to measure gross rates of P cycle processes (Di et al., 2000).**", lines 79-82.

3) Mostly fine. I would just change the sentence "effects of increased Pi mobilization due to microbial lysis on Pi sorption-desorption could be tested in sterile soils". How about deleting "of increased Pi mobilization due to"? I think this leaves what this is about – effects of microbial lysis on Pi sorption-desorption….

Done, the sentence now reads: "**Nonetheless, the effects of increased $P_i$ mobilization due to microbial lysis on $P_i$ sorption-desorption could be tested in sterile soils by adding increasing concentrations of non-labelled $P_i$ alongside the $^{33}P_i$ tracer and then could be corrected for in future $^{33}P$-IPD experiments**" in lines 503-505.

4) Fine.

5) Somehow the text that you give here in the response letter does not completely match the text in the manuscript. And even though you give some references above, I would say that a reference to one of Fardeau's papers would be justified where you speak about extrapolations of r(t)/R.

Done, the sentence now reads: "**Short-term exchange kinetics are then extrapolated over the full time period of the moist soil incubation (E(t)) (Fardeau et al., 1991)**" in line 76-77.

6) Fine

7) Fine

**16: I leave this to readers to think about – the added sentence, however, is certainly useful.**

Thank you.

**22: this should be explained in the manuscript!**

We added this information in line 232-233: "**Time point 0 was assessed by adding the tracer solution and immediately extracting the soils with 0.5 M $NaHCO_3$**".

**24 and 36: not clear if you made any changes to the text?**

We changed the following sentence, and in revision 2 further changed from "seems favorable" to "recommended", see lines 573-575: "**Given the continued extraction of Pi from exchangeable Pi pools in serial extraction tests, parallel determination of microbial P and $^{33}P$ by CFE is recommended compared to sequential extractions by sECE**".

**28: I don't understand the sentence "In these previous studies abiotic processes were not corrected from the final data." And you should add "In the present study," before "This abiotic correction was performed by applying IPD calculations".**

We corrected the text to make the point more clear, see lines 261-270: "**Calculation of gross IPD rates followed the mass balance equations of Kirkham and Bartholomew (1954), as later applied by others for soil gross P fluxes (Kellogg et al., 2006;Mooshammer et al., 2012). In these gross P flux studies abiotic processes were not corrected for, $P_i$ influx rates therefore representing the sum of biotic (organic P mineralization) and abiotic (desorption) processes, the latter of which do not play a significant role in decomposing litter being devoid of soil minerals**

(Mooshammer et al., 2012). However, to calculate gross $P_o$ mineralization for soils, gross rates of $P_i$ desorption have to be corrected for in live soils. In the present study, this abiotic correction was performed by applying IPD calculations for influx (GI, gross influx; equation 1) for sterile soils (abiotic influx by $P_i$ desorption) and live soils (total $P_i$ influx), and taking the difference as biotic influx (i.e. gross $P_o$ mineralization)".

**48: well, you should facilitate reading for your audience…**

We did not change the numbering of the soils.

**50: ok, please add these sentences to the manuscript.**

We added the following information on differences in specific activities between live and sterile soils in lines 337-339: "Specific activities of $P_i$ were initially higher in live than in sterile soils (Fig. 4C). This was caused by the addition of the same amount of radiotracer to both, sterile and live soils, but autoclaving caused a flush of $P_i$ from lysed soil microbes, which effectively lowered the specific activities of $P_i$ in sterile soils".

Once these points have been addressed, the manuscript is acceptable for publication.